# Learning-Augmented Robust Algorithmic Recourse

**Kshitij Kayastha**                                                                    *kk985@drexel.edu*
*Department of Computer Science, Drexel University, USA*

**Vasilis Gkatzelis**                                                                    *gkatz@drexel.edu*
*Department of Computer Science, Drexel University, USA*
Archimedes, Athena Research Center, Greece

**Shahin Jabbari**                                                                    *shahin@drexel.edu*
*Department of Computer Science, Drexel University, USA*

**Reviewed on OpenReview:** *https://openreview.net/forum?id=IFssttzxnP*

## Abstract

Algorithmic recourse provides individuals who receive undesirable outcomes from machine learning systems with minimum-cost improvements to achieve a desirable outcome. However, machine learning models often get updated, so the recourse may not lead to the desired outcome. The robust recourse framework chooses recourses that are less sensitive to adversarial model changes, but this comes at a higher cost. To address this, we initiate the study of learning-augmented algorithmic recourse and evaluate the extent to which a designer equipped with a prediction of the future model can reduce the cost of recourse when the prediction is accurate (consistency) while also limiting the cost even when the prediction is inaccurate (robustness). We propose a novel algorithm, study the robustness-consistency trade-off, and analyze how prediction accuracy affects performance.

## 1 Introduction

Machine learning models are deployed even in sensitive domains such as lending or hiring, e.g., financial institutions use these models to determine whether someone should receive a loan. Given the major impact of these decisions on people's lives, a plethora of work in responsible machine learning aims to make these models fair (Berk et al., 2021; Barocas et al., 2019; Hardt et al., 2016; Zafar et al., 2017), transparent (Lakkaraju et al., 2016; Rudin, 2019), and explainable (Ribeiro et al., 2016; Lundberg and Lee, 2017; Smilkov et al., 2017). A notable line of work along this direction, called *algorithmic recourse* (Wachter et al., 2018; Ustun et al., 2019), provides each individual who received an undesirable label (e.g., one whose loan request was denied) with a minimum-cost improvement to achieve the desired label.

An important weakness of much of the work on algorithmic recourse is the assumption that models are fixed and do not change. In practice, models are periodically updated to reflect changes in data or environment, causing a recourse to become invalid, i.e., following it may not yield a desirable outcome (Dominguez-Olmedo et al., 2022). To alleviate this problem, Upadhyay et al. (2021) proposed a framework that is *robust* to adversarial model changes and provided an algorithm, ROAR, to compute robust recourses. However, guaranteeing robustness in such a setting leads to significantly increased cost (Pawelczyk et al., 2023a).

Our main goal is to design more practical recourse algorithms that balance robustness and cost. To achieve this goal, we move beyond the adversarial setting and leverage the learning-augmented framework (Mitzenmacher and Vassilvitskii, 2020), which has been used in a surge of recent work to overcome the limitations of adversarial (i.e., worst-case) analysis. Specifically, rather than assuming that the designer has no information regarding the model changes, we assume they can at least formulate some predictions regarding these changes. However, these predictions are unreliable, so our goal is to compute recourses that are near-optimal if the

predictions are accurate (consistency), while maintaining good performance even in the worst case, i.e., even when the predictions are highly inaccurate (robustness).

**Our Results** We first adapt the learning-augmented framework and the objectives of consistency and robustness to algorithmic recourse, and we provide a computationally efficient algorithm aiming to optimize these two objectives, as well as their convex combinations. We then formally prove that our algorithm is always optimal for robustness and consistency for any generalized linear model. Note that this is a non-convex problem and, to the best of our knowledge, this is the first optimal algorithm for any robust recourse problem (prior work used algorithms which, as we show, return local rather than global optima). Therefore, this is a contribution to the broader robust recourse literature, beyond the learning-augmented setting. We then experimentally evaluate the algorithm's performance on a variety of both linear and non-linear models.

In our experiments, we use our algorithm to evaluate the achievable trade-offs between consistency and robustness across different datasets, models, and predictions. In all of these settings, our algorithm returns multiple recourses that Pareto dominate ROAR's recourses, i.e., they simultaneously achieve better robustness and consistency. Moving beyond consistency and robustness (which correspond to the extreme cases of perfect predictions and adversarial predictions), we evaluate the performance of our algorithm given near-accurate predictions. Specifically, we evaluate the algorithm's "smoothness" as a function of the prediction error and measure how smoothness depends on the extent to which the algorithm "trusts" the prediction. Finally, we compare our algorithm to prior work for the combinations of validity and cost. Our results indicate that our algorithm achieves higher validity and, in many cases, even without suffering a (much) higher cost.

## 1.1 Related Work

The emerging literature on the interpretability and explainability of machine learning systems mainly advocates for two main approaches. The first approach aims to build inherently simple or interpretable models such as decision lists (Lakkaraju et al., 2016) or generalized additive models (Wang et al., 2017; Yang et al., 2017). These approaches provide *global* explanations for the deployed models. The second approach attempts to explain the decisions of complex black-box models (such as deep neural networks) only on specific inputs (Ribeiro et al., 2016; Lundberg and Lee, 2017; Smilkov et al., 2017; Sundararajan et al., 2017; Selvaraju et al., 2017; Agarwal et al., 2021; Koh and Liang, 2017; Barshan et al., 2020; Ehyaei et al., 2024). These approaches provide a *local* explanation of the model and are sometimes referred to as post-hoc explanations.

Recourse is a post-hoc counterfactual explanation that aims to provide the lowest cost modification that changes the prediction for a given input with an undesirable prediction under the current model (Wachter et al., 2018; Ustun et al., 2019; Looveren and Klaise, 2021; Rawal and Lakkaraju, 2020; Slack et al., 2021; Pawelczyk et al., 2023a; Garg et al., 2025). Since its introduction, different formulations have been used to model the optimization problem in recourse (see (Verma et al., 2020) for an overview). Wachter et al. (2018) and Pawelczyk et al. (2023a) considered score-based classifiers and defined modifications to help instances achieve the desired scores. On the other hand, for binary classifiers, Ustun et al. (2019) required the modification to result in the desired label. Roughly speaking, the first setting can be viewed as a relaxation of the second setting, and we follow the second formulation in our work.

The follow-up works on the problem study several other aspects such as focusing on specific models such as linear models (Ustun et al., 2019) or decision-trees (Kanamori et al., 2024; Bewley et al., 2024), understanding the setting and its implicit assumptions and implications (Barocas et al., 2020; Venkatasubramanian and Alfano, 2020; Fokkema et al., 2024; Gao and Lakkaraju, 2023), attainability or actionability (Joshi et al., 2019; Pawelczyk et al., 2020; Karimi et al., 2020a; Ustun et al., 2019), imperfect causal knowledge (Karimi et al., 2020b), fairness in terms of cost of implementation for different subgroups (Guldogan et al., 2023; Gupta et al., 2019; Heidari et al., 2019), repeated dynamics (Fonseca et al., 2023; Bell et al., 2024; Ehyaei et al., 2024), temporal data (Buliga et al., 2025) and providing model-agnostic explanations for recourse by computing responsiveness scores (Cheon et al., 2025). Extending our work to account for these different aspects of recourse is left for future work.

The most closely related work to us is by Upadhyay et al. (2021), which initiated the study of robust recourse and proposes an algorithm called RObust Algorithmic Recourse (ROAR). We study the same framework for our robustness analysis. Nguyen et al. (2022) proposed a new framework for robust recourse

that uses a different objective than ROAR and focuses on data shifts instead of model shifts. They call their framework *Robust Bayesian Recourse (RBR)* and we refer to their algorithm with the same abbreviation. The RoCourseNet algorithm (Guo et al., 2023) also provided robust recourse, though a direct comparison with this algorithm is not possible, as it is an end-to-end approach, i.e., it simultaneously optimizes for the learned model and robust recourse while the initial model is fixed in our approach, just like in (Upadhyay et al., 2021; Nguyen et al., 2022). Similar to the literature on algorithmic recourse, the literature on robust algorithmic recourse has also studied specialized settings (for different model classes, cost functions, and model changes) and additional desiderata such as fairness (Leofante and Potyka, 2024; Yetukuri et al., 2024; Jiang et al., 2024a; Nguyen et al., 2023; Dutta et al., 2022; Mochaourab et al., 2021). Neither approach uses predictions, though we compare our algorithm with both ROAR and RBR when studying the trade-off between validity and cost of robust recourse in the absence of predictions. Very recently, Turbal et al. (2025) provides provably robust algorithms for recourse under predictive multiplicity (Marx et al., 2020) when the set of models corresponds to an ellipsoidal approximation of the Rashomon set (Breiman, 2001) (i.e., models that are similar in performance on the data distribution (Semenova and Rudin, 2019; Xin et al., 2022)). However, in our setting, the set of adversarial models might perform significantly differently on the data distribution. Even more recently Kyaw et al. (2026) introduced algorithms for robust recourse under linear models and $L^p$-bounded model changes for the case of $p < \infty$ and showed that this restriction on the adversarial model changes can cause the price of recourse to decrease compared to our adversarial model. See (Jiang et al., 2024b) for a recent survey on robust recourse.

In addition to the above works, Pawelczyk et al. (2023b) studies how model updates due to the "right to be forgotten" can affect recourse validity. Dominguez-Olmedo et al. (2022) showed that minimum cost recourse solutions are provably not robust to adversarial perturbations in the model and then presented robust recourse solutions for linear and differentiable models. Black et al. (2022) observe that recourse in deep models can be invalid by small perturbations and suggest that the model's Lipschitzness at the counterfactual point is the key to preserving validity. Hamman et al. (2023) proposed the notion of "naturally-occurring" model change, and provide recourse with theoretical guarantees on the validity of the recourse.

Recourse is also closely related to adversarial training or robust machine learning (Madry et al., 2018; Wong and Kolter, 2018; Athalye et al., 2018). Pawelczyk et al. (2022) studied the connections between various recourse formulations and adversarial training literature. In fact, ROAR (Upadhyay et al., 2021) builds on gradient-based methods originally developed for adversarial training. The convergence of such algorithms has been studied extensively under various assumptions (see, e.g., (Wang et al., 2019)). Theoretical guarantees for adversarial training typically rely on specific data distributions and model classes. For instance, with data drawn from a mixture of Gaussians, the optimal robust classifiers are linear (Li et al., 2020). Moreover, Awasthi et al. (2019) propose an algorithm for learning robust linear classifiers under the realizability assumption and establish hardness results for robust learning of degree-2 polynomial threshold functions. Beyond linear models, theoretical analyses have also been extended to other classes such as decision trees (Vos and Verwer, 2021; 2022). To our knowledge, however, none of these algorithms directly address our setting.

The learning-augmented framework is applied to a wide variety of settings, aiming to provide a refined understanding of the performance guarantees that are achievable beyond the worst case. Unlike approaches in robust optimization that utilize uncertainty sets (Ben-Tal et al., 2009), the learning augmented framework does not impose any assumptions on the quality of the prediction. One of the main application domains is the design of algorithms (e.g., online algorithms (Lykouris and Vassilvitskii, 2018; Purohit et al., 2018)), but it has also been used in data structures design (Kraska et al., 2018), mechanisms interacting with strategic agents (Agrawal et al., 2022; Xu and Lu, 2022), and privacy-preserving methods for processing sensitive data (Khodak et al., 2023). This is already a vast and rapidly growing literature; see (Lindermayr and Megow, 2024) for a frequently updated and organized list of related papers.

## 2 Preliminaries

Consider a predictive model $f_\theta : \mathcal{X} \to \mathcal{Y}$, parameterized by $\theta \in \Theta \subseteq \mathbb{R}^d$, which maps instances (e.g., loan applicants) from a feature space $\mathcal{X} \subseteq \mathbb{R}^d$ to an outcome space $\mathcal{Y} = \{0, 1\}$. The values of 0 and 1 represent undesirable and desirable outcomes (e.g., loan denial or approval), respectively. If a model $f_{\theta_0}$

yields an undesirable outcome for some instance $x_0 \in \mathcal{X}$, i.e., $f_{\theta_0}(x_0) = 0$, the goal is to suggest an *optimal* recourse: the least costly way to modify $x_0$ (how an applicant should strengthen their application) so that the resulting instance $x'$ would achieve the desirable outcome under $f_{\theta_0}$, i.e., $f_{\theta_0}(x') = 1$. Given a cost function $c : \mathcal{X} \times \mathcal{X} \to \mathbb{R}_+$ quantifying the cost $c(x', x_0)$ required to modify $x_0$ to $x'$, the recourse is defined as the following optimization problem (Upadhyay et al., 2021):

$$\min_{x' \in \mathcal{X}} \ell\left(f_{\theta_0}(x'), 1\right) + \lambda \cdot c\left(x', x_0\right), \tag{1}$$

where $\ell : \mathbb{R} \times \mathbb{R} \to \mathbb{R}_+$ is a convex loss function that is decreasing in its first argument (such as binary cross entropy or squared loss) and captures the extent to which the condition $f_{\theta_0}(x') = 1$ is violated and $\lambda \geq 0$ is a regularizer that balances the degree of violation from the desirable outcome and the cost of modifying $x_0$ to $x'$. The regularizer $\lambda$ can be decreased gradually until the desired outcome is reached. In this work, following the approach of (Upadhyay et al., 2021; Rawal and Lakkaraju, 2020), we assume the cost function is the $L^1$ distance, i.e. $c(x, x') = \|x - x'\|_1$.

We denote the *price* of a recourse $x'$ for a model $\theta$ using

$$J(x_0, x', \theta, \lambda) = \ell\left(f_\theta(x'), 1\right) + \lambda \cdot c(x', x_0). \tag{2}$$

For simplicity, we write the price as $J(x', \theta)$ instead.

Equation (1) assumes that the parameters of the model remain the same over time, but in practice, predictive models may be periodically retrained and updated (Upadhyay et al., 2021). These updates can cause a recourse that is valid in the original model (i.e., one that would lead to the desirable outcome in that model) to become invalid in the updated model (Dutta et al., 2022; Black et al., 2022). It is, therefore, natural to require a recourse solution whose validity is robust to (slight) changes in the model parameters.

In line with prior work on robust recourse (Upadhyay et al., 2021; Black et al., 2022), we assume that the parameters of the updated model can be any $\theta' \in \Theta_\alpha$, where $\Theta_\alpha = \{\theta : \|\theta - \theta_0\|_\infty \leq \alpha\} \subseteq \Theta$ is a "neighborhood" around the parameters, $\theta_0$, of the original model, defined using the $L^\infty$ distance and a *known* parameter $\alpha$. Given $\theta_0$ and $\Theta_\alpha$, the *robust* solution would be to choose a recourse $x_r$ that minimizes the price assuming the parameters of the updated model $\theta' \in \Theta_\alpha$ are chosen adversarially, i.e.,

$$x_r \in \arg\min_{x' \in \mathcal{X}} \max_{\theta' \in \Theta_\alpha} J(x', \theta'). \tag{3}$$

## 3  Learning-Augmented Framework

Choosing the robust recourse $x_r$ according to (3) optimizes the price against an adversarially chosen $\theta' \in \Theta_\alpha$, but this price may be much higher than the optimal price if we knew $\theta'$. This is due to the overly pessimistic assumption that the designer has no information regarding the realized $\theta' \in \Theta_\alpha$. To overcome similarly pessimistic results in other domains, the learning-augmented framework (Mitzenmacher and Vassilvitskii, 2020) provides a more refined analysis by assuming the designer is equipped with an *unreliable prediction* and then seeks to achieve near-optimal performance whenever the prediction is accurate while simultaneously maintaining some robustness even if the prediction is arbitrarily inaccurate.

To adapt the learning-augmented framework to algorithmic recourse, we assume that the designer can generate, or is provided with, a possibly unreliable prediction $\hat{\theta} \in \Theta_\alpha$ regarding the model's parameters after the model change. For example, in lending, a prediction can be inferred by information on whether the lender would tighten or loosen its policy over time, or can convey more information in the form of an exact prediction for the future model. If the designer trusts the accuracy of prediction $\hat{\theta}$, then an optimal solution is to choose a recourse $x_c$ consistent with $\hat{\theta}$ i.e.,

$$x_c \in \arg\min_{x' \in \mathcal{X}} J(x', \hat{\theta}). \tag{4}$$

Since the prediction is unreliable, following it blindly can lead to poor performance. In the learning-augmented framework, the *robustness* and *consistency* measures are used to evaluate performance.

**Definition 3.1** (Robustness). Given a parameter $\alpha$, the *robustness* of a recourse $x' \in \mathcal{X}$ is

$$R(x', \alpha) = \max_{\theta' \in \Theta_\alpha} J(x', \theta') - \max_{\theta' \in \Theta_\alpha} J(x_r, \theta'), \tag{5}$$

where $x_r$ is defined in Equation (3).

The robustness evaluates the worst-case price of $x'$ against an adversarial change of the model and then compares it to the price of $x_r$. The robustness is always at least 0 (achieved by $x' = x_r$), and lower is more desirable. While prior work (Upadhyay et al., 2021) measures robustness in absolute terms, we evaluate it relative to $x_r$ to enable a direct comparison between robustness and consistency.

**Definition 3.2** (Consistency). Given a prediction $\hat{\theta} \in \Theta_\alpha$, the consistency of a recourse $x' \in \mathcal{X}$ is

$$C(x', \hat{\theta}) = J(x', \hat{\theta}) - J(x_c, \hat{\theta}), \tag{6}$$

where $x_c$ is defined in Equation (4).

The consistency is always at least zero (achieved by $x' = x_c$), and lower is more desirable.

Choosing $x_r$ guarantees an optimal robustness of 0, but can lead to poor consistency. Choosing $x_c$ guarantees an optimal consistency of 0, but can lead to poor robustness. We study the trade-off between robustness and consistency by studying the following problem:

$$\min_{x'} \beta \cdot R(x', \alpha) + (1 - \beta) \cdot C(x', \hat{\theta}), \tag{7}$$

where $\beta \in [0, 1]$. Solving for varying $\beta$s results in recourses ranging from the optimal robust recourse at $\beta = 1$ to the optimal consistent recourse at $\beta = 0$.

### 3.1 Computing Robust/Consistent Recourses

We next provide Algorithm 1 to optimize Objective (7). We introduce a few additional notations. We use $\text{sgn}(s)$ which is 1 when $s > 0$, 0 when $s = 0$, and $-1$ when $s < 0$. When applied to a vector, the sgn is applied element-wise. For an integer $n \in \mathbb{N}$, $[n] := \{1, \ldots, n\}$.

---

**ALGORITHM 1:** RobustnessConsistencyTradeoff

---

**Input**: $x_0$, $f_{\theta_0}$, $\ell$, $c$, $\alpha$, $\beta$, $\hat{\theta}$  **Output**: $x'$

1: $\theta \leftarrow$ linear approximation of $f_{\theta_0}$ at $x_0$
2: Initialize $x' \leftarrow x_0$ and $\text{ACTIVE}=[d]$
3: **for** $i \in [d]$ **do**
4:     Initialize worst-case $\theta'[i]$ based on $x'[i]$ and $\theta[i]$
5:     $\text{ACTIVE} \leftarrow \text{ACTIVE} \setminus \{i\}$ if $x'_i$ cannot improve objective (7)
6: **while** $\text{ACTIVE} \neq \emptyset$ **do**
7:     $i, \Delta \leftarrow \text{FINDOPTIMALDIMENSIONANDUPDATE}(x_0, \theta', \ell, c, \beta, \hat{\theta}, x')$
8:     **if** $\Delta = 0$ **then**
9:         break                                                 ▷ Terminate
10:     **if** $\text{sgn}(x'[i] + \Delta) = \text{sgn}(x'[i])$ **then**
11:         $x'[i] \leftarrow x'[i] + \Delta$                                 ▷ Update $x'[i]$
12:     **else**
13:         $x'[i] \leftarrow 0$                        ▷ Update but only until it reaches 0
14:         **if** $|\theta[i]| > \alpha$ **then**
15:             $\theta'[i] \leftarrow \theta[i] + \alpha \cdot \text{sgn}(x_0[i])$                    ▷ Modify $\theta'$
16:         **else**
17:             $\text{ACTIVE} \leftarrow \text{ACTIVE} \setminus \{i\}$
18: **return** $x'$

---

Algorithm 1 starts by approximating $f_\theta$ locally at $x_0$ with a linear model e.g., using LIME (Ribeiro et al., 2016). The algorithm then computes the worst-case model $\theta'$ for the "default" recourse of $x' = x_0$ (lines 3-5).

---

**ALGORITHM 2:** FINDOPTIMALDIMENSIONANDUPDATE

---

**Input**: $x_0$, $\theta'$, $\ell$, $c$, $\beta$, $\hat{\theta}$, and $x'$ **Output**: $i, \Delta$

1: $C \leftarrow \vec{0}$, $O \leftarrow -\vec{\infty}$        $\triangleright$ Keep track of change of $x'$ and objective in each dimension
2: **for** $i \in [d]$ **do**
3:   $K(x) = \beta \cdot J(x, \theta') + (1 - \beta) \cdot J(x, \hat{\theta})$    $\triangleright$ Similar to objective (7) without max over $\theta'$ and constants
4:   $x^* \in \arg\min_{x:x[j]=x'[j]\forall j \neq i} K(x)$     $\triangleright$ Minimizer of $K$ if only dimension $i$ could change
5:   **if** $x^*[i] < x'[i] \leq x_0[i]$ or $x^*[i] > x'[i] \geq x_0[i]$ **then**
6:    $O[i] \leftarrow K(x') - K(x^*)$     $\triangleright$ Update if change is consistent with prior changes in $x'[i]$
7:    $C[i] \leftarrow x^*[i] - x'[i]$
8: $i \leftarrow \arg\max_{k \in [d]} O[k], \Delta \leftarrow C[i]$    $\triangleright$ Dimension $i$ which most decreases $K$ and the change $\Delta$ in dimension
9: **return** $i, \Delta$

---

Then, facing $\theta'$, the algorithm greedily modifies $x'$ while simultaneously updating $\theta'$ to ensure that it remains the worst-case model for $x'$ (lines 6-17). More specifically, in each iteration of the while loop, for the current $\theta'$, the algorithm first calls the subroutine FINDOPTIMALDIMENSIONANDUPDATE (line 7), implemented in Algorithm 2. This subroutine searches over each dimension and returns the dimension $i$, and the change $\Delta$ in that dimension that minimizes the objective (7), assuming $\theta'$ and all other coordinates of $x'$ are fixed. Suppose the $\Delta$ returned by Algorithm 2 is 0, meaning no improvement can be made on any dimension, then Algorithm 1 terminates (line 9). If $|\Delta| > 0$, but applying this change to $x'[i]$ does not cause $x'[i]$ to change sign (i.e., $x'[i] + \Delta$ has the same sign as $x'[i]$), then the update is applied; in that case, the adversarial model $\theta_0$ does not need to be updated (line 11). On the other hand, if the recommended change flips the sign of $x'[i]$, this would cause the adversarial response to change as well. The algorithm instead applies this change all the way up to $x'[i] = 0$ (i.e., up until the sign change), which maintains the same adversarial response. Then, $\theta'$ is updated with $\theta'[i] \leftarrow \theta[i] + \alpha \cdot \text{sgn}(x_0[i])$ (lines 13-15), so that any subsequent changes of $x'[i]$ beyond $x'[i] = 0$ are calculated with respect to the correct adversarial response. If, this update of $\theta'$ causes the sign of $\theta'[i]$ to change (i.e., if $|\theta[i]| > \alpha$), then no further change in this dimension can yield an improvement on our objective and $i$ is removed from the ACTIVE set (line 17).

Algorithm 1 is efficient since (1) the linear approximation of $f_{\theta_0}$ can be computed in polynomial time (Ribeiro et al., 2016), and (2) the while loop runs for $O(d)$ iterations and the running time of each iteration is dominated by the runtime of subroutine FINDOPTIMALDIMENSIONANDUPDATE. This subroutine runs in polynomial time (e.g., by running at most $d$ one-dimensional gradient descents (Boyd and Vandenberghe, 2014)). See Appendix B for more details, and Appendix C.7 for the scalability experiments.

We analyze the theoretical properties of Algorithm 1 by focusing on generalized linear models. A model $f_\theta$ is generalized linear if $f_\theta(x) := g \circ h_\theta(x)$, where $h_\theta : \mathcal{X} \to \mathbb{R}$ is a linear function and $g : \mathbb{R} \to [0, 1]$ is a non-decreasing function mapping the outputs of $h_\theta$ to probabilities e.g., setting $g$ to the sigmoid recovers logistic regression. Our main result is as follows.

**Theorem 3.3.** *Suppose $f_\theta$ is a generalized linear model, $\beta \in \{0, 1\}$, and the single-dimensional convex optimization in line 4 of Algorithm 2 can be solved in polynomial time. Then Algorithm 1 returns a recourse $x' \in \arg\min_x \beta R(x, \alpha) + (1 - \beta)C(x, \hat{\theta})$ in polynomial time.*

*Proof Sketch.* We provide a sketch of the proof for the case of $\beta = 1$ (i.e., for optimal robustness), and defer the full proof and the analysis of the case of $\beta = 0$ (i.e., for optimal consistency) to Appendix B.2.

We start with Observation B.2, which points out that we can assume $J(x', \theta')$ is a function only of the cost $\|x' - x_0\|_1$ of $x'$ and its inner product with its worst-case (adversarial) model, $x' \cdot \theta'$. Specifically, it is a linear increasing function of the former and a convex decreasing function of the latter. Using this observation, we view the problem as choosing a recourse $x'$ and suffering a cost of $\|x' - x_0\|_1$, aiming to maximize $x' \cdot \theta'$, and an adversary then chooses $\theta'$ aiming to minimize this inner product. The actual form of $J(\cdot)$ determines the extent to which we may want to trade off the cost of recourse $x'$ for an increase in the inner product, but the convexity with respect to the inner product suggests a decreasing marginal gain with respect to the latter.

To better understand the structure of the adversarial response, Lemma B.3 shows that for any given recourse $x$, the adversarial model $\theta'$ is essentially equal to $\theta_0 - \alpha \cdot \text{sgn}(x)$. In other words, the adversary shifts each

coordinate $i$ by $\alpha$ (the maximum shift that is allowed while remaining within $\Theta_\alpha$), and the direction is determined by whether $x[i]$ is positive or negative. As a result, we can assume that the adversarial choice of $\theta'$ for each dimension $i$ is always either $\theta_0 + \alpha$ or $\theta_0 - \alpha$. This also allows us to partition $\mathcal{X}$, the space of all possible recourses, into regions that would face the same adversary. Specifically, if $x$ and $x'$ are such that $\text{sgn}(x) = \text{sgn}(x')$, then their worst-case model is the same.

Our algorithm starts from $x_0$ and gradually changes it until it reaches the robust recourse $x_r$. To determine its first step, the algorithm computes the worst-case model $\theta'$ for $x_0$ using the formula provided by Lemma B.3. If we were to assume that the adversary would remain fixed at $\theta'$ no matter how we change the recourse, then computing the optimal recourse would be easy: we would identify the dimension $i$ with the largest $|\theta'[i]|$ value, and we would change only this dimension of $x_0$ (This would in turn simplify how $i$ in line 7 can be computed for the specific case of $\beta = 1$. See Appendix B.2). This is optimal because changing some other dimension $j$ by $\Delta$ would increase our cost by $\Delta$ and increase the inner product by $\Delta \cdot |\theta'[j]|$ against a fixed adversary. How much we should change that dimension would then be determined by solving the optimization problem in line 7. However, if this change "flipped" the sign of $x'[i]$, this would also change the adversary, potentially compromising the optimality of $x'$.

Our proof shows that a globally optimal recourse can be computed by myopically optimizing $x'$ until the adversary needs to be updated, and then repeating the same process until the marginal gain in the inner product is outweighed by the marginal increase in cost. This is partly due to the fact (shown in Lemma B.5) that the order in which this myopic approach considers the dimension of the recourse to change is optimal, exhibiting a decreasing sequence of $|\theta'[i]|$ values. $\qquad\square$

Note that $\beta = 1$ corresponds to computing the optimal robust recourse. Even for generalized linear models, the objective function in Equation (7) is non-convex when $\beta = 1$ (see Appendix B.1). Hence, prior gradient-based approaches (Upadhyay et al., 2021; Nguyen et al., 2022) cannot guarantee optimality. To the best of our knowledge, Algorithm 1 is the first optimal algorithm for any robust recourse formulation. This optimality guarantee relies on the assumption that the single-dimensional convex problem in line 4 of Algorithm 2 can be solved in polynomial time. This can be done, for example, when the loss function $\ell$ is known by deriving first-order conditions and verifying optimality conditions. In general, gradient-based approaches can be used to converge to an optimal solution, in which case the solution provided by Algorithm 1 will also converge to the optimal robust or consistent recourse.

The idea of approximating a non-linear function locally has also been used in prior work (Upadhyay et al., 2021; Rawal and Lakkaraju, 2020). In Section 4, we empirically compare the performance of our algorithm to prior work in linear and non-linear settings. To handle feasibility constraints or categorical features, the recourse of Algorithm 1 can be post-processed (e.g., by projection) to guarantee feasibility (Upadhyay et al., 2021; Nguyen et al., 2022; Guo et al., 2023). See Appendix C.4.

## 4 Experiments

In this section, we describe the datasets, implementation details, and present our findings. Our code and data are available at `https://github.com/kshitij-kayastha/LearningAugmentedRobustRecourse`.

**Datasets** We experiment on synthetic and real-world data. For the synthetic dataset, we follow Upadhyay et al. (2021) to generate 1000 data points in 2-d. For each data point, we first sample a label $y \in \{0, 1\}$ uniformly at random. We then sample the instance corresponding to this label from a Gaussian distribution $\mathcal{N}(\mu_y, \Sigma_y)$. We set $\mu_0 = [-2, -2]$, $\mu_1 = [+2, +2]$, and $\Sigma_0 = \Sigma_1 = 0.5\mathbb{I}$. See Figure 1(a) in (Upadhyay et al., 2021). We also use two real datasets: (a) The German Credit dataset (Hofmann, 1994), which consists of 1000 data points, each with 7 features containing information about a loan applicant (age, marital status, income, and credit duration), and binary labels good (1) or bad (0) determine the creditworthiness. (b) The Small Business Administration dataset (Min Li and Taylor, 2018), which contains the small business loans approved by the State of California from 1989 to 2004. The dataset includes 1159 data points, each with 28 features containing information about the business (business category, zip code, and number of jobs created by the business), and the binary labels indicate whether the small business has defaulted on the loan (0) or

not (1). For real-world data, we normalize the features. We use the datasets to learn the initial model $\theta_0$. See Appendix C.7 for experiments on a larger dataset.

**Implementation Details** We use 5-fold cross-validation in all experiments: 4 folds to train the initial model $\theta_0$ and the remaining fold to compute recourse. Recourse is only computed for instances with label (0) under $\theta_0$, and we report averages over folds and test instances in all experiments. We used logistic regression as our linear model and trained it using Scikit-Learn. As our non-linear model, we used a 3-layer neural network with 50, 100, and 200 nodes in each successive layer (same as (Upadhyay et al., 2021)). The network uses ReLU activation functions, binary cross-entropy loss, and Adam optimizer, and is trained for 100 epochs using PyTorch. See Appendix C.1 for additional implementation details.

To generate recourse, we implemented Algorithm 1 with different $\beta$ values. For non-linear models, we first approximate them locally with LIME (Ribeiro et al., 2016) (same as (Upadhyay et al., 2021)). We used the code from (Upadhyay et al., 2021) and (Nguyen et al., 2022) for ROAR and RBR's implementation as baselines. We next describe our parameter choices. We use binary cross-entropy as the loss function $\ell$ and $L^1$ distance as the cost function $c$. We measure closeness using the $L^\infty$ norm for model parameters. In Section 4.1, we follow the same procedure as in (Upadhyay et al., 2021) for selecting $\alpha$ and $\lambda$. We fix an $\alpha$ and greedily search for $\lambda$ that maximizes the recourse validity under the original model $\theta_0$. We study the effect of varying $\alpha$ and $\lambda$ in Section 4.2 and Appendix C.3. In each experiment, we specify how $\hat{\theta}$ is selected. See our code and Appendix C.1 for more details.

## 4.1 Findings: Learning-Augmented Setting

**Robustness-Consistency Trade-off** To study the trade-off between consistency and robustness, we generated 5 predictions. For logistic regression models, we generated 4 perturbations by adding or subtracting $\alpha$ in each dimension of $\theta_0$. For neural network models, we added the perturbation to the LIME approximation of $f_{\theta_0}$. Along with $\theta_0$, these form the 5 model parameters we used as predictions. We use $\alpha = 0.5$ for the trade-off results (see Appendix C.3 for different $\alpha$s). For each given prediction $\hat{\theta}$, we run Algorithm 1 and a modification of ROAR (Upadhyay et al., 2021) as a baseline for varying $\beta \in [0, 1]$ and compute the robustness and consistency of the solution using Equations (5) and (6).

In Figure 1, the rows and columns correspond to different datasets and models. In each sub-figure, curves show the robustness-consistency trade-off of recourses by Algorithm 1 for different predictions (indicated by different colors) and ROAR (same color for each prediction by with lines that include $\star$.

For each curve, the bottom right point corresponds to $\beta = 1$ (the optimal robust recourse). Given the optimality of Algorithm 1 for linear models, the robustness is 0, though empirically our algorithm achieves near 0 robustness for non-linear models too. However, these recourses might have different consistencies depending on the prediction. Similarly, the top left point of each curve corresponds to $\beta = 0$ (the optimal consistent recourse) with a consistency of 0, which might have different robustness. ROAR does not achieve either optimal robustness or optimal consistency.[1]

Our algorithm Pareto dominates ROAR, simultaneously achieving better robustness and consistency for all $\beta$ values. Furthermore, at $\beta = 1$, the robustness of ROAR's recourses is substantially worse compared to ours (e.g., ROAR achieves a robustness of 17.64 on the neural network models learned on synthetic data (Figure 1b)). ROAR is also much slower than our algorithm (See Section 4.2). We generally observe that the poor consistency of the robust solutions can be decreased substantially with a small increase in robustness. This is also true for decreasing the robustness of the consistent solutions, albeit to a smaller degree.

**Smoothness** In this section, we study how prediction error can affect the recourse quality. In particular, for each dataset-model pair, we first compute the *correct prediction* $\hat{\theta}_*$, corresponding to the realized future model (by either shifting the data or temporal changes in data collection (Upadhyay et al., 2021)). We then generate incorrect predictions by perturbing each coordinate of the correct prediction by different values in

---

[1]The optimization problem in Equation 7 is convex but non-differentiable at $\beta = 0$ for linear models. Hence, gradient-based methods with fixed step sizes, such as ROAR, are not guaranteed to find the optimal solution.

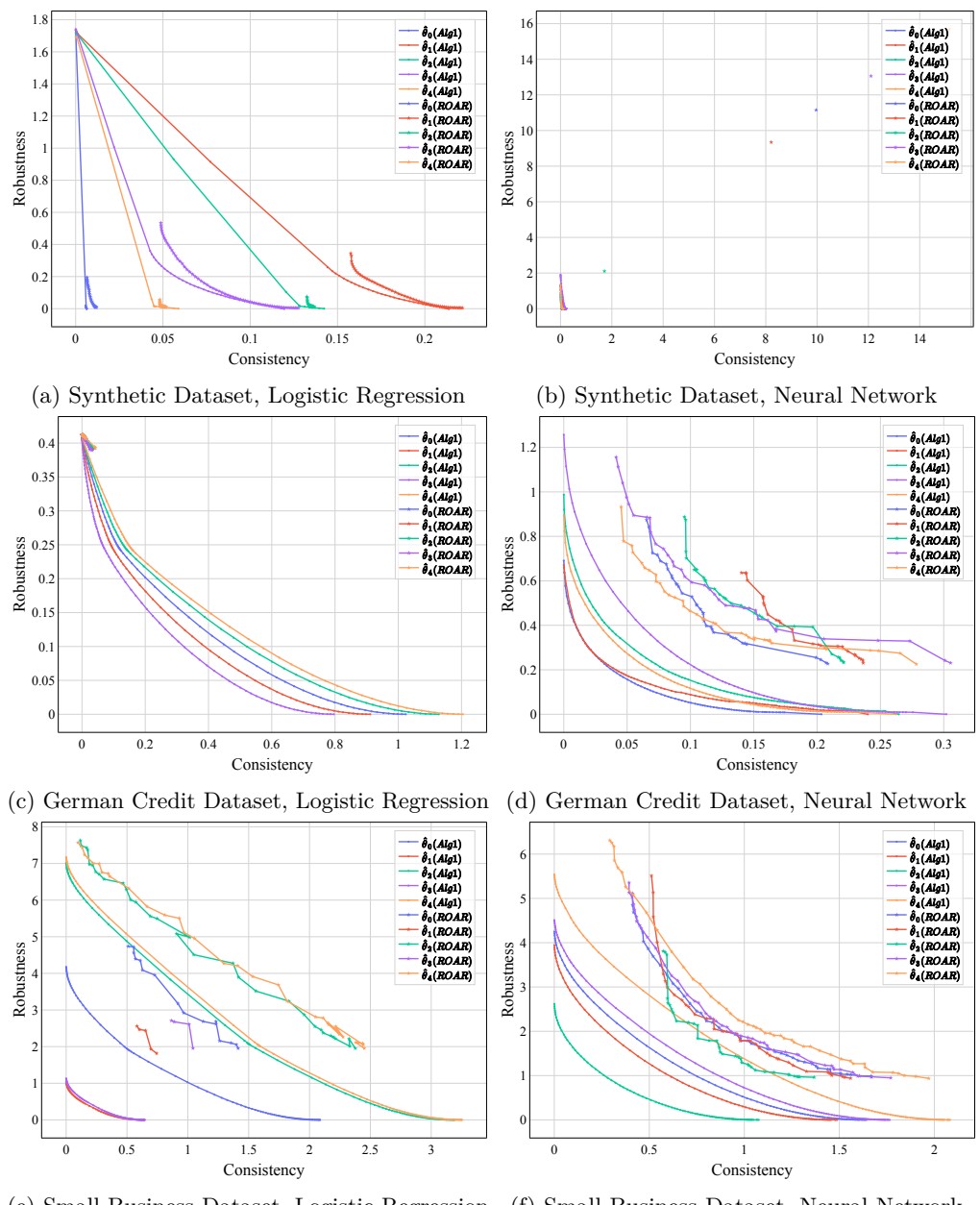

Figure 1: The trade-off between robustness and consistency for $\alpha = 0.5$: rows and columns correspond to different datasets and models as indicated in the sub-caption. In each subfigure, each curve shows the trade-off for different predictions for our algorithm and ROAR.

$\{+\epsilon, -\epsilon, +2\epsilon, -2\epsilon\}$, corresponding to the amount of error. The $\epsilon$ values depend on the dataset and model and are chosen to ensure that all the predictions are in $\Theta_\alpha$ for $\alpha = 1$.

Given a prediction $\hat{\theta}$ and a parameter $\beta \in [0, 1]$, Algorithm 1 generates a recourse. If $\beta = 1$, it ignores the prediction, and if $\beta = 0$ it fully trusts it; values of $\beta \in (0, 1)$ strike a balance between these two extremes. To measure the performance as a function of the prediction error we define a metric called *smoothness*: $J(x'(\beta, \hat{\theta}), \hat{\theta}_*) - J(x'(\hat{\theta}_*), \hat{\theta}_*)$, where $x'(\beta, \hat{\theta})$ is the computed recourse, $\hat{\theta}_*$ is the correct prediction and $x'(\hat{\theta}_*)$ is the consistent recourse for the correct prediction. The smoothness is non-negative and it is 0 if the learner is

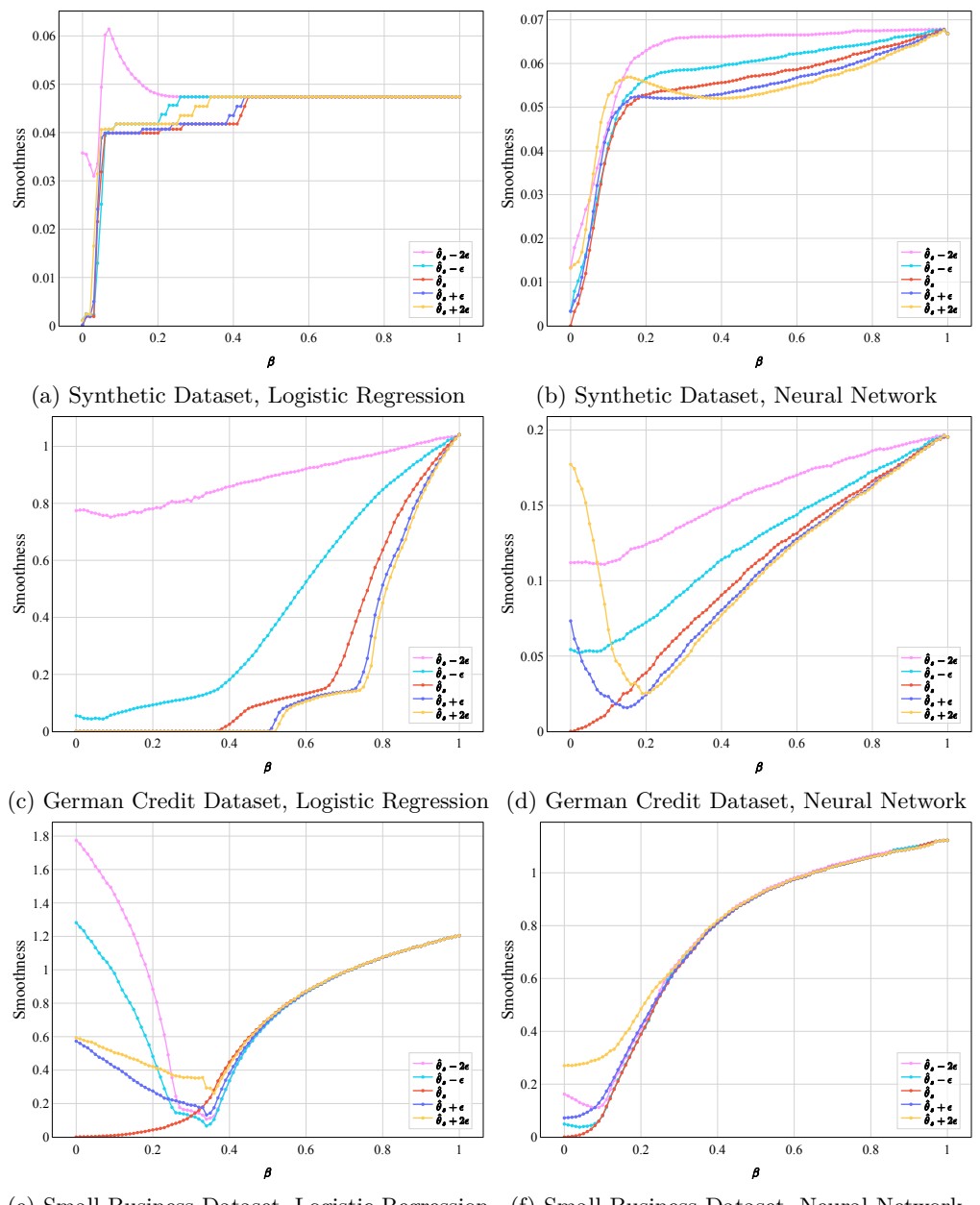

Figure 2: The smoothness for predictions with different accuracies: rows and columns correspond to different datasets and models as indicated in the sub-caption. In each subfigure, curves correspond to different predictions and track the smoothness as a function of $\beta$ for the given prediction.

provided with the correct prediction ($\hat{\theta} = \hat{\theta}_*$) and fully trusts it ($\beta = 0$). Lower smoothness values correspond to better performance despite the error.

In Figure 2, the rows and columns correspond to different datasets and models. In each sub-figure, curves show the smoothness of Algorithm 1 for different predictions as a function of $\beta$. There are 5 lines in each subfigure corresponding to the correct prediction and the perturbations. Focusing on $\beta = 0$, we observe that smoothness increases sharply as a function of the prediction "error", since the algorithm fully trusts the (incorrect) prediction. As $\beta \to 1$, the smoothness of all predictions converges to the same value, since the algorithm disregards them. In some cases, this convergence occurs at smaller $\beta$ values (Figure 2e),

but other cases require $\beta$ to be very close to 1 (Figure 2a). While the smoothness monotonically increases with $\beta$ when using the correct prediction, using incorrect predictions results in interesting non-monotone behavior and even leads to improved performance compared to using the correct prediction (Figure 2a). In Appendix C.5, we provided baselines using a variant of ROAR, though the generated recourses generally have higher smoothness.

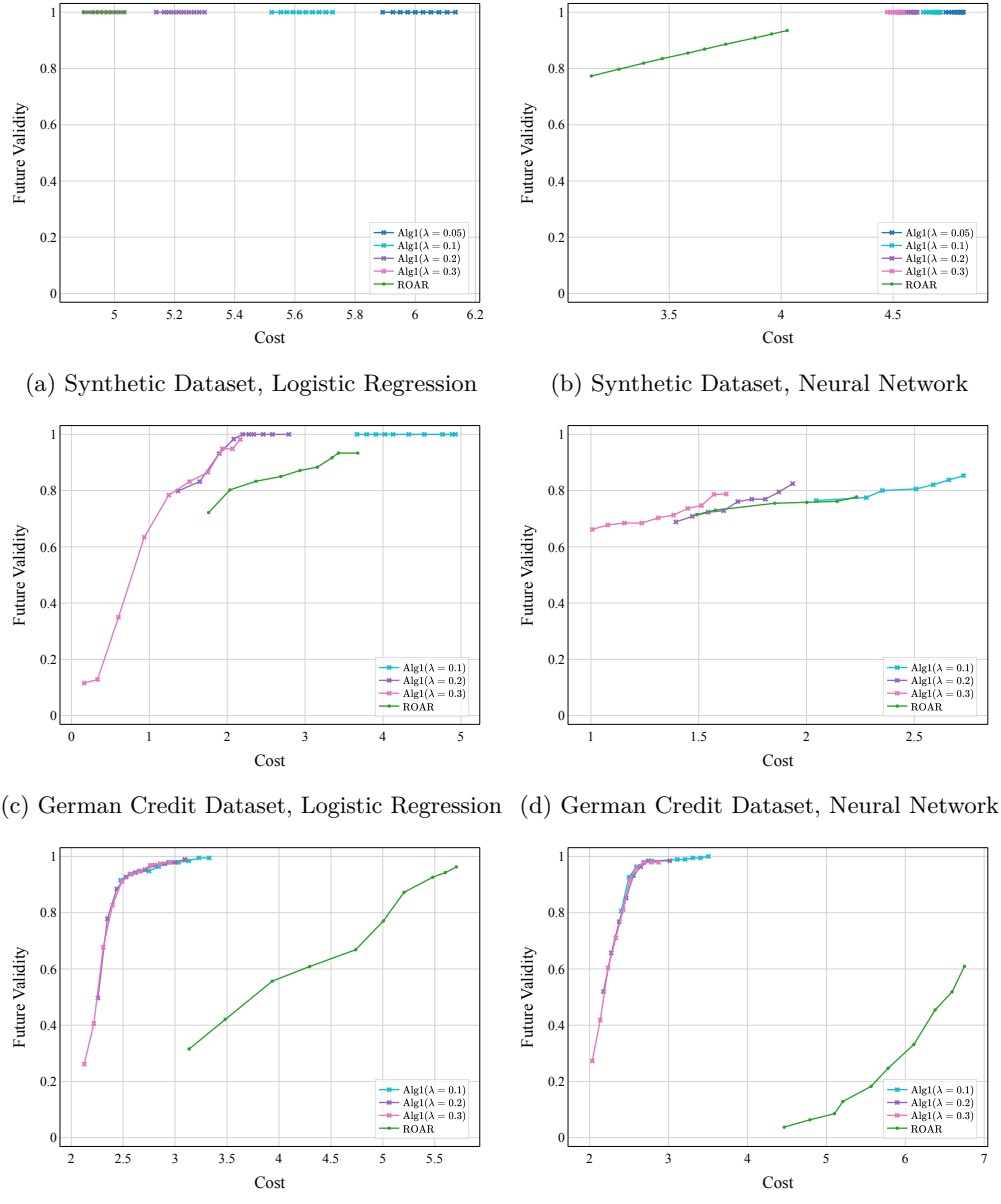

(a) Synthetic Dataset, Logistic Regression     (b) Synthetic Dataset, Neural Network

(c) German Credit Dataset, Logistic Regression     (d) German Credit Dataset, Neural Network

(e) Small Business Dataset, Logistic Regression     (f) Small Business Dataset, Neural Network

Figure 3: The trade-off between future validity and cost: rows and columns correspond to different datasets and models. In each subfigure, curves show the trade-off for different algorithms.

## 4.2 Findings: Computing Robust Recourse

By setting $\beta = 1$, our algorithm can be used to compute robust recourse. In this section, we perform a more detailed comparison with ROAR by breaking down the robustness to understand the effect of each term in

Equation (1). The first term is a proxy for *validity* and the second term is the cost of modifying $x_0$. To compute validity, as opposed to computing a possibly different model for each instance, as is done till now, we compute a single model for the entire dataset. Following prior work (Upadhyay et al., 2021; Nguyen et al., 2022), this model is obtained by training a model on a shifted version of the datasets. We call the validity with respect to this new model as *future validity*. See Appendix C.1.

Figure 3 depicts the trade-off between future validity and cost for all datasets and models. In each curve, we varied $\alpha \in [0.02, 0.2]$ in increments of 0.02. We used 4 different $\lambda$s: 0.05, 0.1, 0.2, and 0.3, and the trade-off for each $\lambda$ is plotted with a different color. To avoid overcrowding, we only included the results of $\lambda = 0.1$ for ROAR. Using $\lambda = 0.05$ does not change the trade-off for ROAR, and using $\lambda = 0.2$, or 0.3 degrades the validity even further. For our algorithm, sometimes different $\lambda$ values create similar trade-offs, in which case we only include the results for one of them. Also note that our algorithm runs much faster (10-100x) than ROAR for all parameters. See Appendix C.1.

Figure 3 indicates that our algorithm often Pareto dominates ROAR, i.e., it achieves higher validity with a smaller cost. While the validity of our algorithm often approaches 1 for smaller $\lambda$ values, this is not the case for ROAR. In addition, the validity is lower for both approaches when using neural network models. Moreover, consistent with Pawelczyk et al. (2023a)'s observation, the cost of recourse can increase significantly for validity values close to 1.

Finally, the LIME approximation can be unreliable, affecting the quality of the trade-off generated by our algorithm. In Appendix C.6, we provide further comparisons with RBR (Nguyen et al., 2022), an approach designed to address this shortcoming. We observe that our algorithm performs similarly to or better than RBR even when the quality of approximation is not perfect, and only when this quality is extremely low (German dataset) does its performance become dominated by RBR. Even in such cases, our algorithm still achieves a high validity (while requiring a higher cost). See Appendix C.6 for more details.

## 5 Conclusion and Discussion

The robust recourse literature has focused on many aspects: different model classes, costs, model changes, and optimization approaches resulting in formulations that require different solutions (see (Jiang et al., 2024a)). Furthermore, adopting the learning-augmented framework introduces additional modeling aspects, such as prediction, definitions of robustness and consistency, and measures of their trade-off. We initiated the study of the learning-augmented robust recourse and followed the assumptions in the closest prior work to allow for a direct comparison. We highlight several extensions.

First, while our framework can handle customizable weights for different inputs, using any norm as the cost function implies that the features can be modified independently. Some prior work considers these dependencies and the *actionability* of recourse (Karimi et al., 2020a; Joshi et al., 2019; Pawelczyk et al., 2020). We leave these as future work. Second, our notion of robustness and consistency measures the performance of the algorithm against optimal solutions in an additive manner, similar to *regret* (Cesa-Bianchi and Lugosi, 2006). This comparison can also be made multiplicatively, similar to *competitive ratio* (Mitzenmacher and Vassilvitskii, 2020), which we leave as future work. Third, as is common in the learning-augmented framework (Agarwal et al., 2021; Mitzenmacher and Vassilvitskii, 2020), we assumed the prediction about the updated model is explicitly given. A natural way to compute such a prediction, in practice, is through performativity (Perdomo et al., 2020). For example, the implementation of recourse can cause the distribution shift, and the designer can use this knowledge to form a prediction for the anticipated future model (König et al., 2025). However, this prediction can be imperfect since individuals might not exactly implement the recourse (Fonseca et al., 2023). Moreover, in practice, the feedback might be *weaker* or even *noisy* (Bechavod et al., 2022). Incorporating these into our framework is an exciting future work direction. We also leave a potential theoretical analysis or study new robustness frameworks to future work.

## Acknowledgment

We thank Kaidi Xu and Phone Kyaw for insightful discussions. Vasilis Gkatzelis was partially supported by the NSF CAREER award CCF-2047907 and the NSF grant CCF-2210502.

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

# A ROAR

For completeness, we provide the details of ROAR (Upadhyay et al., 2021) in this section. ROAR is inspired by the vast literature on adversarial training (see e.g., (Madry et al., 2018) and utilizes Danskin's Theorem (Danskin, 1967)) to compute the gradients (with respect to $x'$). The pseudocode is provided in Algorithm 3. ROAR requires the function $J$ to be differentiable for $x'$. Furthermore, ROAR relies on the ability to compute a maximizer of $J$ for $\theta'$. When $J$ is differentiable for $\theta'$, a local maximum can be computed with projected gradient ascent.

---

**ALGORITHM 3:** RObust Algorithmic Recourse (ROAR)

---

**Input**: $x_0$, $\theta_0$, $\ell$, $c$, $\alpha$, $L^p$, $\eta$ (learning rate)
**Output**: $x'$
1: Initialize $x' \leftarrow x_0$.        ▷ Initialization for the robust solution
2: Initialize $g \leftarrow \vec{0}$.        ▷ Initialization for gradients
3: **Repeat**
4:      $\theta' \leftarrow \arg\max_{\theta : \|\theta - \theta_0\|_p \leq \alpha} J(x', \theta)$        ▷ Maximizer of $J$ with respect to $\theta$ for the current $x'$
5:      $g \leftarrow \nabla_{x'} J(x', \theta')$        ▷ The gradient of $J$ with respect to $x'$ for the current $\theta'$
6:      $x' \leftarrow x' - \alpha g$        ▷ A gradient descent step to update $x'$
7: **Until** convergence
8: **return** $x'$.

---

# B Omitted Details from Section 3.1

## B.1 Non-convexity of Robust Recourse for Linear Models

**Proposition B.1.** *Suppose $f$ is a generalized linear model, $c$ is the $L^1$ norm, and $\ell(.,1)$ is a convex loss function that is decreasing in its first argument. Then, there exist choices for $\ell$ and $x_0$ such that the objective in Equation 3 is non-convex in $x$.*

*Proof.* Note that at $\beta = 1$, Equation (7) is the same as Equation (5). Furthermore, the set of solutions to Equation 5 and Equation (3) is identical since the only difference between the two equations is the additional second term in Equation (5), which is a constant.

We provide a concrete example that makes it easy to verify the non-convexity of the optimization problem in Equation (3) even for linear models. Consider an instance in one dimension where $x_0 = [1, 1]$ (note that the second dimension is the unchangeable intercept), $\theta_0 = [0, 0]$, $\ell$ is squared loss, $\alpha = 0.5$, and $\lambda = 1$. For any recourse, $x_r = [x, 1]$ (note that the intercept cannot change), the worst-case $\theta'$ is of the form $[0.5\text{sgn}(x), -0.5]$ since $\alpha$ is 0.5 and $\theta_0$ is 0 in both dimensions. The cost of recourse for $x_r$ can be written as $1/\left(e^{0.5x\text{sgn}(x)-0.5}\right)^2 + |x - 1|$. Plotting this one-dimensional function proves that this function is not convex. □

## B.2 Proof of Theorem 3.3

To prove Theorem 3.3 and verify the optimality of Algorithm 1 for generalized linear models and $\beta \in \{0, 1\}$, we first make some observations and prove some useful lemmas.

First of all, at $\beta = 1$, Equation (7) is the same as Equation 5. Furthermore, the set of solutions to Equation (5) and Equation (3) is identical since the only difference between the two equations is the additional second term in Equation (5), which is a constant. Moreover, at $\beta = 0$, Equation (7) is the same as Equation (6). Furthermore, the set of solutions to Equation 6 and Equation (3) with $\alpha = 0$ is identical since the only difference between the two equations at $\alpha = 0$ is that the second term is a constant. Hence, to prove Theorem 3.3, it suffices to show that Algorithm 1 will compute the optimal solution for Equation (3) when $f_{\theta_0}$ is a generalized linear model i.e., the $x'$ returned by Algorithm 1 satisfies $x' \in \arg\min_{x \in \mathcal{X}} \max_{\theta' \in \Theta_\alpha} J(x, \theta')$.

To simplify the exposition, we rewrite a simplified version of Algorithm 1 for this specific case of $\beta = 1$ and generalized linear models and flesh out all the omitted details. We present this simplification in Algorithm 4. The summary of the simplifying steps is as follows: (1) The pre-processing step (lines 3-5 in Algorithm 1) are expanded to distinguish between when $x_0$ is initially 0 at any coordinate or not. If $x_0[i]$ is 0 in any coordinate and the adversarial $\theta$ can change sign (i.e., $\theta_0[i] < \alpha$), the optimal choice for $x'[i]$ would be 0. Instead of waiting for line 17 in Algorithm 1 to detect this, the simplified implementation removes dimension $i$ from the ACTIVE set to improve the running time. (2) Instead of calling the subroutine FINDOPTIMALDIMENSIONANDUPDATE (line 7 of Algorithm 1), the simplified implementation computes the dimension $i$ by finding the dimension in the ACTIVE set that have the highest $\theta'$ value since any change of a fixed amount provides the most bang-per-buck in that dimension (line 12). Then line 13 of Algorithm 4 computes an identical calculation to line 3 in Algorithm 1 to compute the best change $\Delta$ in the dimension that needs to be updated. (3) And finally, Algorithm 4 does not check for $\Delta$ being 0 and terminates when the update in one dimension can be done without changing the sign (line 16). This is simply to speed up the running time, since, as we show in the proof, without termination, $\Delta$ would be 0 in the next iteration.

So to prove Theorem 3.3, it suffices to show that Algorithm 1 will compute the optimal solution for Equation (3) when $f_{\theta_0}$ is a generalized linear model. Without loss of generality, throughout this section, we will be assuming that $\theta_0[i] \neq \theta_0[j]$ for any two dimensions $i \neq j$. This can be easily guaranteed by an arbitrarily small perturbation of these values without having any non-trivial impact on the model, but all of our results hold even without this assumption; it would just introduce some requirement for tie-breaking that would make the arguments slightly more tedious. Furthermore, we use $e_i$ to denote a $d$-dimensional unit vector with all zeros except for the $i$-th coordinate, which has a value of one.

---

**ALGORITHM 4:** Detailed Description of Algorithm 1 for $\beta = 1$ and generalized linear model $f_{\theta_0}$

**Input** : $x_0$, $\theta_0$, $\ell$, $c$, $\alpha$
**Output**: $x'$
 1: Initialize $x' \leftarrow x_0$
 2: Initialize ACTIVE=$[d]$            ▷ Set of coordinates to update
 3: **for** $i \in [d]$ **do**
 4:    **if** $x_0[i] \neq 0$ **then**
 5:       Initialize $\theta'[i] \leftarrow \theta_0[i] - \alpha \cdot \text{sgn}(x_0[i])$     ▷ Initialization for $\theta'$ (the worst-case model)
 6:    **else**
 7:       **if** $|\theta_0[i]| > \alpha$ **then**
 8:          Initialize $\theta'[i] \leftarrow \theta_0[i] - \alpha \cdot \text{sgn}(\theta_0[i])$
 9:       **else**
10:          ACTIVE $\leftarrow$ ACTIVE $\setminus \{i\}$     ▷ Remove the coordinate that cannot improve $J$
11: **while** ACTIVE $\neq \emptyset$ **do**
12:    $i \leftarrow \arg\max_{j \in \text{ACTIVE}} |\theta'[j]|$     ▷ Next coordinate to update
13:    $\Delta \leftarrow \arg\min_{\Delta} J(x' + \Delta e_i, \theta') - J(x', \theta')$     ▷ Compute the best update for the selected coordinate
14:    **if** $\text{sgn}(x'[i] + \Delta) = \text{sgn}(x'[i])$ **then**
15:       $x'[i] \leftarrow x'[i] + \Delta$     ▷ Apply the update and terminate
16:       break
17:    **else**
18:       $x'[i] \leftarrow 0$     ▷ Update the coordinate but only until it reaches 0
19:       **if** $|\theta_0[i]| > \alpha$ **then**
20:          $\theta'[i] \leftarrow \theta_0[i] + \alpha \cdot \text{sgn}(x_0[i])$     ▷ Modify $\theta'$ accordingly
21:       **else**
22:          ACTIVE $\leftarrow$ ACTIVE $\setminus \{i\}$
23: **return** $x'$

---

**Observation B.2.** For a fixed set of parameter values, the problem of optimizing robustness in our setting can be captured as computing a recourse $x_r$ aiming to minimize the value of a function $J(\cdot)$ whose value depends only on the distance cost of $x'$, i.e., $\|x' - x_0\|_1$, and its inner product with an adversarially chosen $\theta' \in \Theta_\alpha$. Formally, our goal is to compute a recourse $x_r$ such that:

$$x_r \in \arg\min_{x' \in \mathcal{X}} \max_{\theta' \in \Theta_\alpha} J(\|x' - x_0\|_1, \; x' \cdot \theta').$$

Also, $J(\cdot)$ is a linear increasing function of $\|x' - x_0\|_1$ and a convex decreasing function of $x' \cdot \theta'$.

Observation B.2 provides an alternative interpretation of the problem: by choosing a recourse $x'$, we suffer a cost $\|x' - x_0\|_1$ and the adversary then chooses a $\theta'$ aiming to minimize the value of the inner product $x' \cdot \theta'$. This implies that for a given $x'$, a choice of $\theta'$ is not optimal for the adversary unless it minimizes this inner product. Also, it implies that among all choices of $x'$ with the same cost $\|x' - x_0\|_1$, the optimal one has to maximize the inner product $x' \cdot \theta'$ with the adversarially chosen $\theta'$. We use this fact to prove that a recourse $x'$ is not a robust choice by providing an alternative recourse with the same cost and a greater dot product.

Our first lemma provides additional structure regarding the optimal adversarial choice in response to any given recourse $x$.

**Lemma B.3.** *For any recourse $x$, the adversarial response $\theta' = \arg\max_{\theta \in \Theta_\alpha} J(x, \theta)$ is such that $\theta'[i] = \theta_0[i] + \alpha$ for each dimension $i$ such that $x[i] < 0$ and $\theta'[i] = \theta_0[i] - \alpha$ for each dimension $i$ such that $x[i] > 0$. For any dimension $i$ with $x[i] = 0$ we can without loss of generality assume that $\theta'[i] \in \{|\theta_0[i] + \alpha|, |\theta_0[i] - \alpha|\}$.*

*Proof.* For any dimension $i$ with $x[i] = 0$, it is easy to verify that no matter what the value of $\theta'[i]$ is, the contribution of $x[i] \cdot \theta'[i]$ to the inner product $x \cdot \theta'$ is zero, so we can indeed without loss of generality assume that $\theta'[i] \in \{|\theta_0[i] + \alpha|, |\theta_0[i] - \alpha|\}$. Now, assume that $x[i] < 0$, yet $\theta'[i] < \theta_0[i] + \alpha$, and consider an alternative response $\theta''$ such that $\theta''[i] = \theta_0[i] + \alpha$ and $\theta''[j] = \theta'[j]$ for all other dimensions $j \neq i$. Clearly, $\theta'' \in \Theta_\alpha$, since $|\theta''[i] - \theta_0[i]| = \alpha$ and $|\theta''[j] - \theta_0[j]| \leq \alpha$ for all other dimensions $j \neq i$ as well, by the fact that $\theta' \in \Theta_\alpha$. Therefore, it suffices to prove that $x \cdot \theta'' < x \cdot \theta'$, as this would contradict the fact that $\theta' = \arg\max_{\theta \in \Theta_\alpha} J(x, \theta)$. To verify that this is indeed the case, note that

$$\begin{aligned} x \cdot \theta' - x \cdot \theta'' &= x[i] \cdot \theta'[i] - x[i] \cdot \theta''[i] \\ &= x[i] \cdot (\theta'[i] - \theta''[i]) \\ &> 0, \end{aligned}$$

where the first equation use the fact that $\theta'$ and $\theta''$ are identical for all dimensions except $i$ and the inequality uses the fact that $x[i] < 0$ and $\theta'[i] < \theta''[i]$. A symmetric argument can be used to show that $\theta'[i] = \theta_0[i] - \alpha$ for each dimension $i$ such that $x[i] > 0$. $\qquad \square$

Lemma B.3 shows that for any recourse $x$, an adversarial response that minimizes $x \cdot \theta'$ is $\theta' = \theta_0 - \alpha \cdot \mathrm{sgn}(x)$. Our next lemma shows how the adversarial response to the initial point $x_0$, (i.e., $\theta_0 - \alpha \cdot \mathrm{sgn}(x_0)$) determines the direction toward which each dimension of $x_0$ should be changed (if at all).

**Lemma B.4.** *For any optimal recourse $x_r \in \arg\min_{x' \in \mathcal{X}} \max_{\theta' \in \Theta_\alpha} J(x', \theta')$ and every coordinate $i$, it must be that $x_r$ raises the value of the $i$-th dimension only if the adversary's best response to its original value is positive, and it lowers it only if the adversary's best response to its original value is negative. Using Lemma B.3, we can formally define this as:*

$$\begin{aligned} x_r[i] > x_0[i] \quad &only\ if \quad \theta_0[i] - \alpha \cdot \mathrm{sgn}(x_0[i]) > 0 \\ x_r[i] < x_0[i] \quad &only\ if \quad \theta_0[i] - \alpha \cdot \mathrm{sgn}(x_0[i]) < 0. \end{aligned}$$

*Proof.* Assume that for some dimension $i$ we have $x_r[i] > x_0[i]$ even though $\theta_0[i] - \alpha \cdot \mathrm{sgn}(x_0[i]) < 0$, and let $x'$ be the recourse such that $x'[i] = x_0[i]$ while $x'[j] = x_r[j]$ for all other coordinates, $j \neq i$. If $\theta^*$ is the adversary's best response to $x_r$ and $\theta'$ is the adversary's best response to $x'$, then the difference between the inner product of $x' \cdot \theta'$ and $x_r \cdot \theta^*$ is:

$$\begin{aligned} x' \cdot \theta' - x_r \cdot \theta^* &= x'[i] \cdot \theta'[i] - x_r[i] \cdot \theta^*[i] \\ &= x_0[i] \cdot (\theta_0[i] - \alpha \cdot \mathrm{sgn}(x_0[i])) - x_r[i] \cdot \theta^*[i] \\ &\geq x_0[i] \cdot (\theta_0[i] - \alpha \cdot \mathrm{sgn}(x_0[i])) - x_r[i] \cdot (\theta_0[i] - \alpha \cdot \mathrm{sgn}(x_0[i])) \\ &= (x_0[i] - x_r[i]) \cdot (\theta_0[i] - \alpha \cdot \mathrm{sgn}(x_0[i])) \\ &> 0, \end{aligned}$$

where the first equation uses the fact that $x'[j] = x_r[j]$ for all $j \neq i$, the second equation uses the fact that $x'[i] = x_0[i]$ and the fact that the adversary's best response to $x_0[i]$ is $\theta_0[i] - \alpha \cdot \text{sgn}(x_0[i])$, and the subsequent inequality uses the fact that the product $x_r[i] \cdot \theta^*[i]$ is at most $x_r[i] \cdot (\theta_0[i] - \alpha \cdot \text{sgn}(x_0[i]))$ since the adversary's goal is to minimize this product and adversary's best response to $x_r[i]$ will do at least as well as the best response to $x_0[i]$ (which is a feasible, even if sub-optimal, response for the adversary).

We have shown that the inner product achieved by $x'$ would be greater than that of $x_r$, while the cost of $x'$ is also strictly less than $x_r$, since $x'$ keeps the $i$-th coordinate unchanged. Therefore, $\max_{\theta' \in \Theta_\alpha} J(x', \theta') < \max_{\theta' \in \Theta_\alpha} J(x_r, \theta')$, contradicting the assumption that $x_r \in \arg\min_{x' \in \mathcal{X}} \max_{\theta' \in \Theta_\alpha} J(x', \theta')$. A symmetric argument leads to a contradiction if we assume that $x_r[i] < x_0[i]$ even though $\theta_0[i] - \alpha \cdot \text{sgn}(x_0[i]) > 0$. $\square$

We now prove a lemma regarding the sequence of $|\theta'[i]|$ values of the dimensions that the while loop of Algorithms 4 changes.

**Lemma B.5.** *Let $j_k$ denote the dimension chosen in line 12 of Algorithm 4 during the $k$-th execution of its while-loop, and let $v_k$ denote the value of $|\theta'[j_k]|$ at a point in time (note that $\theta'$ changes over time). The sequence of $v_k$ values are decreasing with $k$.*

*Proof.* Note that in the $k$-th iteration of the while-loop, line 12 of Algorithm 4 chooses $j_k$ so that $j_k = \arg\max_{j \in \text{ACTIVE}} |\theta'[j]|$, based on the values of $\theta'$ at the beginning of that iteration. As a result, if $\theta'$ remains the same throughout the execution of the algorithm (which would happen if $\text{sgn}(x_r) = \text{sgn}(x_0)$, i.e., if none of the recourse coordinates changes from positive to negative or vice versa), then the lemma is clearly true. On the other hand, if the recourse "flips signs" for some dimension $i$, i.e., $\text{sgn}(x_r[i]) \neq \text{sgn}(x_0[i])$, this could lead to a change of the value of $\theta'[i]$. Specifically, as shown in Lemma B.3 and implemented in line 20 of the algorithm, the adversary changes $\theta'[i]$ to $\theta_0[i] + \alpha \cdot \text{sgn}(x_0[i])$. If that transition causes the sign of $\theta'[i]$ to change, then dimension $i$ becomes inactive and the algorithm will not consider it again in the future. If the sign of $\theta'[i]$ remains the same, then we can show that its absolute value would drop after this change, so even if it is considered in the future, it would still satisfy the claim of this lemma. To verify that its absolute value drops, assume that $x_0[i] > 0$, suggesting that the algorithm has so far lowered its value to 0, which would only happen if $\theta_0[i] < 0$ (otherwise, this change would be decreasing the inner product). Since $x_0[i] > 0$, the new value of $\theta'[i]$ is equal to $\theta_0[i] + \alpha$, and since this remains negative, like $\theta_0[i]$, we conclude that its absolute value decreased. A symmetric argument can be used for the case where $x_0[i] < 0$. $\square$

We are now ready to prove our main theoretical result (the proof of Theorem 3.3), showing that Algorithm 4 always returns an optimal robust recourse.

*Proof of Theorem 3.3.* To prove the optimality of the recourse $x_r$ returned by Algorithm 4, i.e., the fact that $x_r \in \arg\min_{x' \in \mathcal{X}} \max_{\theta' \in \Theta_\alpha} J(x', \theta')$, we assume that this is false, i.e., that there exists some other recourse $x^* \in \arg\min_{x' \in \mathcal{X}} \max_{\theta' \in \Theta_\alpha} J(x', \theta')$ such that $\max_{\theta' \in \Theta_\alpha} J(x^*, \theta') < \max_{\theta' \in \Theta_\alpha} J(x_r, \theta')$, and we prove that this leads to a contradiction.

Note that since $x^* \in \arg\min_{x' \in \mathcal{X}} \max_{\theta' \in \Theta_\alpha} J(x', \theta')$, it must satisfy Lemma B.4. Also, note that the way that Algorithm 4 generates $x_r$ also satisfies the conditions of Lemma B.4 (the choice of $\Delta$ in line 13 would never lead to a recourse of higher cost without improving the inner product), so we can conclude that if $x^*$ and $x_0$ were to change the same coordinate they would both do so in the same direction, i.e.,

$$\text{sgn}(x^*[i] - x_0[i]) = \text{sgn}(x_r[i] - x_0[i]).$$

Having established that for every coordinate $i$ the values of $x^*[i]$ and $x_r[i]$ will either both be at most $x_0[i]$ or both be at least $x_0[i]$, the rest of the proof performs a case analysis by comparing how far from $x_0[i]$ each one of them moves:

- **Case 1:** $\|x^* - x_0\|_1 = \|x_r - x_0\|_1$. Since $x^* \neq x_r$, it must be that $|x^*[i] - x_0[i]| > |x_r[i] - x_0[i]|$ for some $i$ and $|x^*[j] - x_0[j]| < |x_r[j] - x_0[j]|$ for some $j$. To get a contradiction for this case as well, we will consider an alternative recourse $x'$ that is identical to $x^*$ except for dimensions $i$ and $j$, each

of which is moved $\delta$ closer to the values of $x_r[i]$ and $x_r[j]$, respectively, for some arbitrarily small constant $\delta > 0$. Formally,

$$x'[i] = x^*[i] + \delta \cdot \text{sgn}(x_r[i] - x^*[i]) \qquad \text{and} \qquad x'[j] = x^*[j] + \delta \cdot \text{sgn}(x_r[j] - x^*[j]).$$

Note that $x^*$ and $x'$ both have the same price since they only differ in $i$ and $j$ and

$$|x^*[i] - x_0[i]| + |x^*[j] - x_0[j]| = |x'[i] - x_0[i]| + \delta + |x'[j] - x_0[j]| - \delta$$
$$= |x'[i] - x_0[i]| + |x'[j] - x_0[j]|.$$

We let $\delta$ be small enough so that the adversary's response to $x^*$ and $x'$ is the same; for this to hold it is sufficient that a value of $x^*$ that is strictly positive does not become strictly negative in $x'$, or vice versa. If we let $\theta'$ denote this adversary, then we have

$$x' \cdot \theta' - x^* \cdot \theta' = |(x'[j] - x^*[j]) \cdot \theta'[j]| - |(x'[i] - x^*[i]) \cdot \theta'[i]|$$
$$= \delta \cdot |\theta'[j]| - \delta \cdot |\theta'[i]|$$
$$= \delta \cdot (|\theta'[j]| - |\theta'[i]|),$$

where the first equality uses the fact that $x^*$ and $x'$ differ only on $i$ and $j$, and the fact that if we replace recourse $x^*$ with $x'$, then the change of $\delta$ on the $j$-th coordinate increases the distance from $x_0[j]$ and thus increases the inner product, while the change of $\delta$ on the $i$-th coordinate decreases the distance from $x_0[i]$ and thus decreases the inner product. The second equality uses the fact that the change on both coordinates $i$ and $j$ is equal to $\delta$.

To conclude with a contradiction, it suffices to show that $|\theta'[j]| > |\theta'[i]|$, as this would imply $x' \cdot \theta' > x^* \cdot \theta'$, contradicting the fact that $x^* \in \arg\min_{x' \in \mathcal{X}} \max_{\theta' \in \Theta_\alpha} J(x', \theta')$, since $x'$ would require the same cost as $x^*$ but it would yield a greater inner product. We consider three possible scenarios: *i)* If Algorithm 4 in line 12 chose to change dimension $i$ facing adversary $\theta'[i]$ before considering dimension $j$ and adversary $\theta'[j]$, then the fact that $|x^*[i] - x_0[i]| > |x_r[i] - x_0[i]|$ implies that the algorithm did not change coordinate $i$ as much as $x^*$ and it must have terminated after that via line 16; this would suggest that dimension $j$ and adversary $\theta[j]$ would never be reached after that, contradicting the fact that $|x^*[j] - x_0[j]| < |x_r[j] - x_0[j]|$. *ii)* If Algorithm 4 in line 12 chose to change dimension $j$ facing adversary $\theta'[j]$ and later on also considered dimension $i$ and adversary $\theta'[i]$, then Lemma B.5 suggests that $|\theta'[j]| > |\theta'[i]|$, once again leading to a contradiction. Finally, *iii)* if Algorithm 4 in line 12 chose to change dimension $j$ facing adversary $\theta'[j]$ and never ended up considering dimension $i$ even though $|\theta'[j]| < |\theta'[i]|$, this suggests that $i$ was removed from the ACTIVE set during the execution of the algorithm, which implies that $x_r[i] = 0$ and $|\theta_0[i]| < \alpha$, so moving further away from $x_0[i]$ would actually hurt the inner product because the adversary can flip the sign of $\theta'[i]$ via a change of $\alpha$. The fact that $x^*$ actually moved dimension $i$ further away then again contradicts the fact that $x^* \in \arg\min_{x' \in \mathcal{X}} \max_{\theta' \in \Theta_\alpha} J(x', \theta')$.

- **Case 2:** $\|x^* - x_0\|_1 < \|x_r - x_0\|_1$. In this case, we can infer that for some $i$ we have $|x^*[i] - x_0[i]| < |x_r[i] - x_0[i]|$, i.e., $x^*$ determined that the increase of the inner product achieved by moving $x^*[i]$ further away from $x_0[i]$ and closer to $x_r[i]$ was not worth the cost suffered by this increase. However, note that as we discussed in Observation B.2, $J(\cdot)$ is a decreasing function of the inner product. Also note that, since Algorithm 4 changes a coordinate of the recourse only if it increases the inner product, there must be some point in time during the execution of the algorithm when the inner product of $x' \cdot \theta'$ was at least as high as the inner product of $x^*$ with the adversarial response to $x^*$. Nevertheless, line 13 determined that this change would decrease the objective value $J(\cdot)$. If we specifically consider the last dimension $j$ changed by the algorithm, using Lemma B.5, we can infer that the value of $|\theta'[j]|$ at the time of this change was less than the value of $|\theta'[i]|$ for the dimension $i$ satisfying $|x^*[i] - x_0[i]| < |x_r[i] - x_0[i]|$; this is due to the fact that the algorithm chose to change $i$ weakly earlier than $j$. As a result, since line 13 determined that the increase of cost was outweighed by the increase in the inner product even though $|\theta'[j]| \leq |\theta'[i]|$, the inner product is greater, and $J(\cdot)$ is convex in the latter, this implies that increasing the value of $x^*[i]$ would also decrease the objective, thus leading to a contradiction of the fact that $x^* \in \arg\min_{x' \in \mathcal{X}} \max_{\theta' \in \Theta_\alpha} J(x', \theta')$.

- **Case 3:** $\|x^* - x_0\|_1 > \|x_r - x_0\|_1$. This case is similar to the one above, but rather than arguing that $x^*$ missed out on further changes that would have led to an additional decrease of the objective, we instead argue that $x^*$ went too far with the changes it made. Specifically, there must be some $i$ such that $|x^*[i] - x_0[i]| > |x_r[i] - x_0[i]|$, i.e., $x^*$ determined that the increase of the inner product achieved by moving $x^*[i]$ further away from $x_0[i]$ than $x_r[i]$ did was worth the cost suffered by this increase. Since the cost of $x^*$ is greater than the cost of $x_r$, it must be the case that its inner product is greater. Therefore, line 13 of the algorithm determined that moving $x_r[i]$ further away from $x_0[i]$ would not lead to an improvement of the objective even for a smaller inner product. Once again, the convexity of $J(\cdot)$ with respect to the inner product combined with the aforementioned facts implies that this increase must have hurt $x^*$ as well, leading to a contradiction. $\square$

## C  Omitted Details from Section 4

In this section, we provide additional results and analysis that were omitted from Section 4 due to space constraints. In Section C.1, we provide additional details on the running time of our algorithm as well as what values were chosen for some of the hyperparameters. Section C.2 provides error bars for the robustness consistency tradeoffs generated by our algorithm. Section C.3 details how parameter changes affect our results. Section C.4 studies the effect of post-processing and actionability of the generated recourse on performance. In Section C.5, we provide baselines for our smoothness analysis in Section 4.1. Section C.6 studies the trade-off between cost and validity when there is uncertainty in the value of $\alpha$. Finally, Section C.7 provides additional results for a larger dataset.

| Model | Dataset | $\lambda$ |
|---|---|---|
| LR | Synthetic Data | 1.0 |
| | German Credit Data | 0.5 - 0.7 |
| | Small Business Data | 1.0 |
| NN | Synthetic Data | 1.0 |
| | German Credit Data | 0.1 - 0.2 |
| | Small Business Data | 1.0 |

Table 1: $\lambda$ that maximize the validity with respect to the original model $\theta_0$ for each dataset. The other choices of parameters are mentioned in Section C.1.

### C.1  Additional Experimental Details

The experiments were conducted on two laptops: an Apple M1 Pro and a 2.2 GHz 6-Core Intel Core i7. Average runtime (across datasets/model) to generate a robust recourse for Algorithm 1, ROAR (Upadhyay et al., 2021), and RBR (Nguyen et al., 2022) are 0.001, 1,0, and 40 seconds, respectively. The total runtime of Algorithm 1 to generate Figure 1 was 45-60 minutes, depending on the subfigure.

In our robustness versus consistency experiments in Section 4.1, we choose $\alpha = 0.5$ and find the $\lambda$ that maximizes the validity with respect to the original model $\theta_0$ in each round of cross-validation. The range of $\lambda$ values found to maximize the $\theta_0$ validity for each setting is reported in Table 1.

In our experiment on smoothness in Section 4.1, we created the future model using a modified dataset similar to (Upadhyay et al., 2021). To produce the altered synthetic data, we employed the same method outlined in Section 4, but we changed the mean of the Gaussian distribution for class 0. The new distribution is $x \sim N(\mu_0', \Sigma_y)$, where $\mu_0'$ is equal to $\mu_0 + [\alpha, 0]^T$, while $\mu_1'$ remained unchanged at $\mu_1$. We used this new distribution to learn a model for the correct prediction. The German credit dataset (Hofmann, 1994) is available in two versions, with the second one (Grömping, 2020) fixing coding errors found in the first. This dataset exemplifies a shift due to data correction. We used the second dataset to learn the model for the

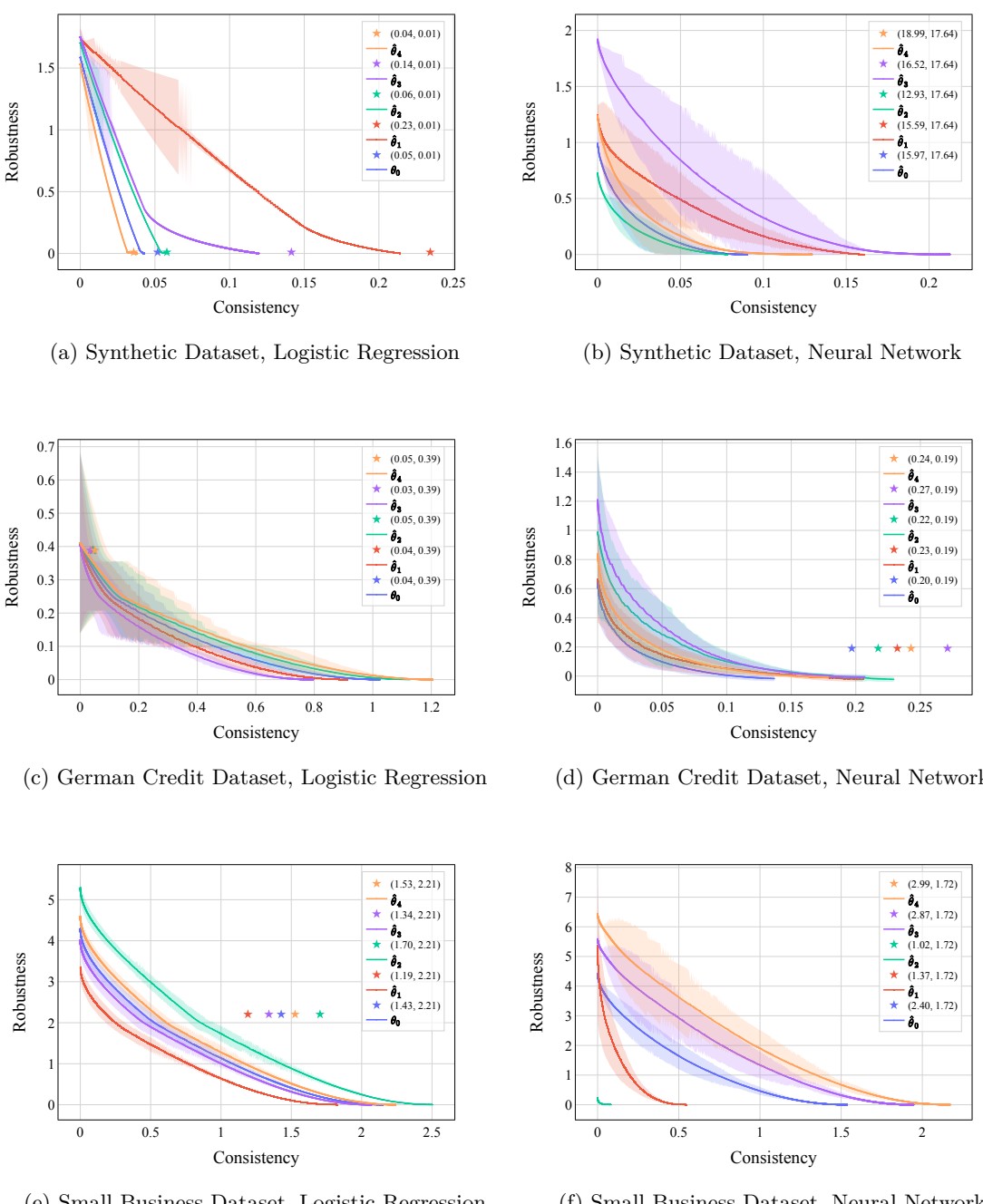

(a) Synthetic Dataset, Logistic Regression

(b) Synthetic Dataset, Neural Network

(c) German Credit Dataset, Logistic Regression

(d) German Credit Dataset, Neural Network

(e) Small Business Dataset, Logistic Regression

(f) Small Business Dataset, Neural Network

Figure 4: The trade-off between robustness and consistency for $\alpha = 0.5$ with error bars for robustness: logistic regression (left) and neural network (right). Rows correspond to datasets: synthetic (top), German (middle), and Small Business (bottom). In each subfigure, each curve shows the trade-off for different predictions. The robustness and consistency of ROAR solutions at $\beta = 1$ are mentioned in parentheses and depicted by stars. Missing stars are outside the range of the figure.

correct prediction. The Small Business Administration dataset (Min Li and Taylor, 2018), which contains data on 2,102 small business loans approved in California from 1989 to 2012, demonstrates temporal shifts.

We split this dataset into two parts: data points before 2006 form the original dataset, while those from 2006 onwards constitute the shifted dataset. We used the shifted dataset to learn a model for the correct prediction.

To generate the predictions in our smoothness experiment in Section 4.1, we define $\epsilon$ as half the distance between the original model $\theta_0$, and the shifted model $\hat{\theta}_*$, which we use as the correct prediction for the future model. Perturbations of $\pm\epsilon$ and $\pm 2\epsilon$ are then applied to each dimension of the $\hat{\theta}_*$. For linear models, we use $\epsilon = 0.12$ for the Synthetic dataset, $\epsilon = 0.16$ for the German dataset, and $\epsilon = 0.43$ for the Small Business Administration dataset. For non-linear models, the amount of perturbation is determined by each instance in the dataset by using the LIME approximation to provide recourse. More details can be found in our code. In all cases, the perturbed values are clamped to ensure they remain within the $\alpha = 1$ in terms of $L^1$ distance from the original model $\theta_0$.

In our cost versus worst-case validity experiments in Section 4.2, motivated by reasons for data shift and given access to different versions of the datasets, we follow prior work (Upadhyay et al., 2021; Nguyen et al., 2022) and compute a model on a shifted version of the dataset and measure validity against this new model. For the synthetic dataset, the shifted data is achieved by changing the mean and variance in the data distribution. For the German Credit dataset, the data shift is due to data correction. For the Small Business Administration dataset, the shift is temporal. See (Upadhyay et al., 2021) for more details.

## C.2 Error Bars for Robustness-Consistency Trade-off

This section provides additional details about the experiments regarding the trade-off between robustness and consistency. In particular, we replicate the performance of Algorithm 1, which is used to generate Figure 1, but also include error bars. These error bars are calculated for the robustness (Figure 4) and consistency costs (Figure 5) when averaging is done over all the data points in the test set that require recourse as well as the folds.

## C.3 Effect of the Parameters

In the experiments on the trade-off between robustness and consistency in Section 4.1, we used $\alpha = 0.5$ and a $\lambda$ that maximizes the validity of recourse with respect to this $\alpha$. In this section, we see how varying $\alpha$ can affect the results. In particular, in Figures 6 and 7, we replicated the trade-offs of our algorithm, which is presented in Figure 1 in Section 4.1 with $\alpha = 0.1$ and $\alpha = 1$, respectively. Again, for each choice of $\alpha$, we selected a $\lambda$ that maximizes the validity of recourse with respect to this $\alpha$. We generally observe that increasing $\alpha$ increases both the robustness and consistency costs.

## C.4 Actionable Features and Post-processing

Similar to prior work (Upadhyay et al., 2021; Nguyen et al., 2022), our algorithm does not directly handle categorical features, and the recourse provided by our algorithm can violate potential constraints or restrictions that might exist on some features. In this section, we study how the performance of our algorithm can be affected under these situations.

In particular, in this section, we repeat the experiments on robustness-consistency trade-off from Section 4.1 but post-process the recourses generated by Algorithm 1 to satisfy categorical and actionable features. In particular, we first remove all non-actionable or immutable features (Guldogan et al., 2023) from the dataset, compute the recourses using Algorithm 1, and then project the recourse to the feasible region to satisfy the constraints posed by categorical features.

Since the synthetic dataset does not include any categorical or immutable features, we exclude it from our analysis. The robustness-consistency trade-off for both logistic regression and neural network models on the German Credit dataset is depicted in Figure 8. Comparing the left panel with no post-processing to the right panel with post-processing, we observe that the achievable trade-off after post-processing becomes slightly worse (e.g., consistency values do not reach 0 in Figure 8d), but the degradation in performance is generally very small.

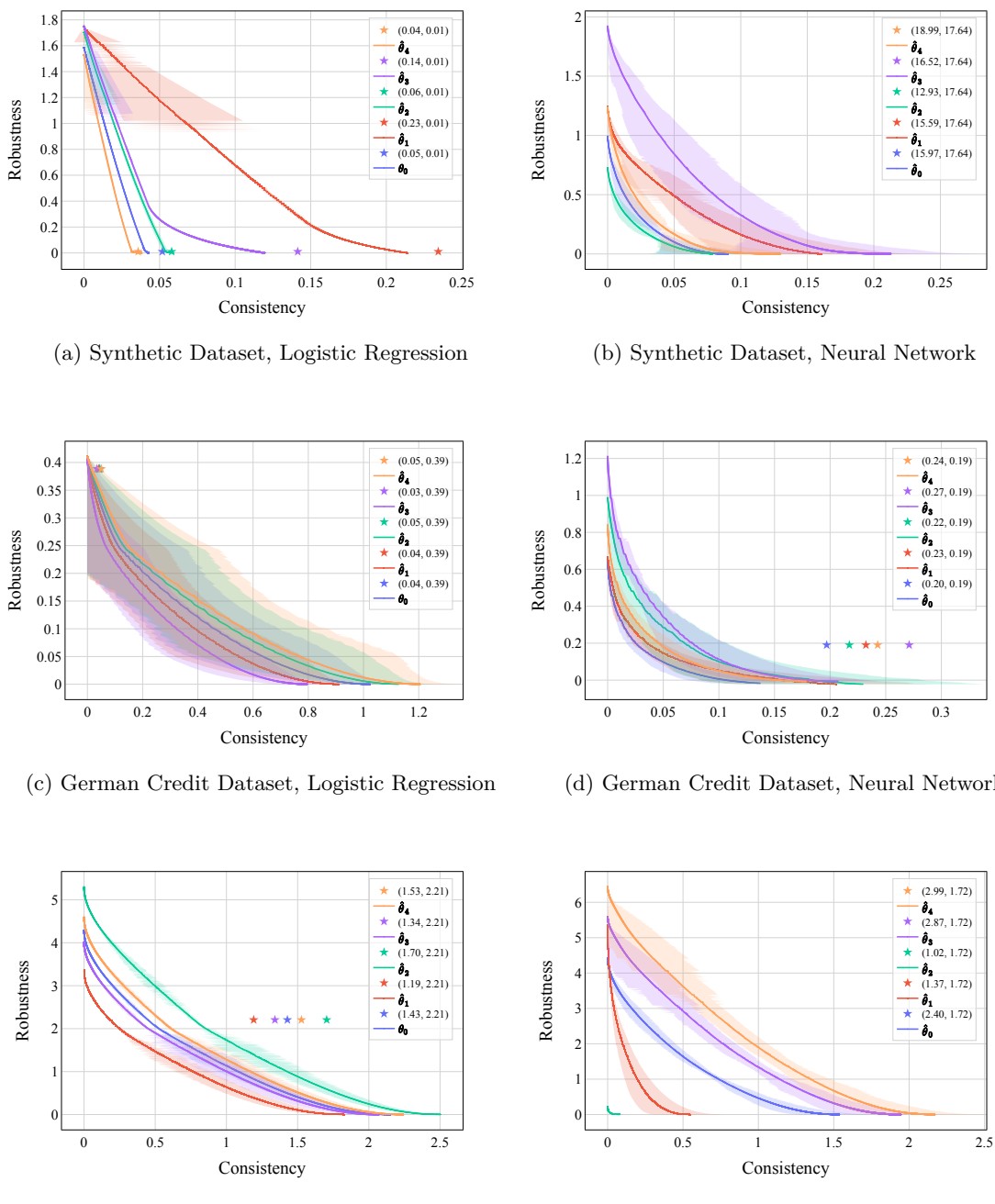

Figure 5: The trade-off between robustness and consistency for $\alpha = 0.5$ with error bars for consistency: logistic regression (left) and neural network (right). Rows correspond to datasets: synthetic (top), German (middle), and Small Business (bottom). In each subfigure, each curve shows the trade-off for different predictions. The robustness and consistency of ROAR solutions at $\beta = 1$ are mentioned in parentheses and depicted by stars. Missing stars are outside the range of the figure.

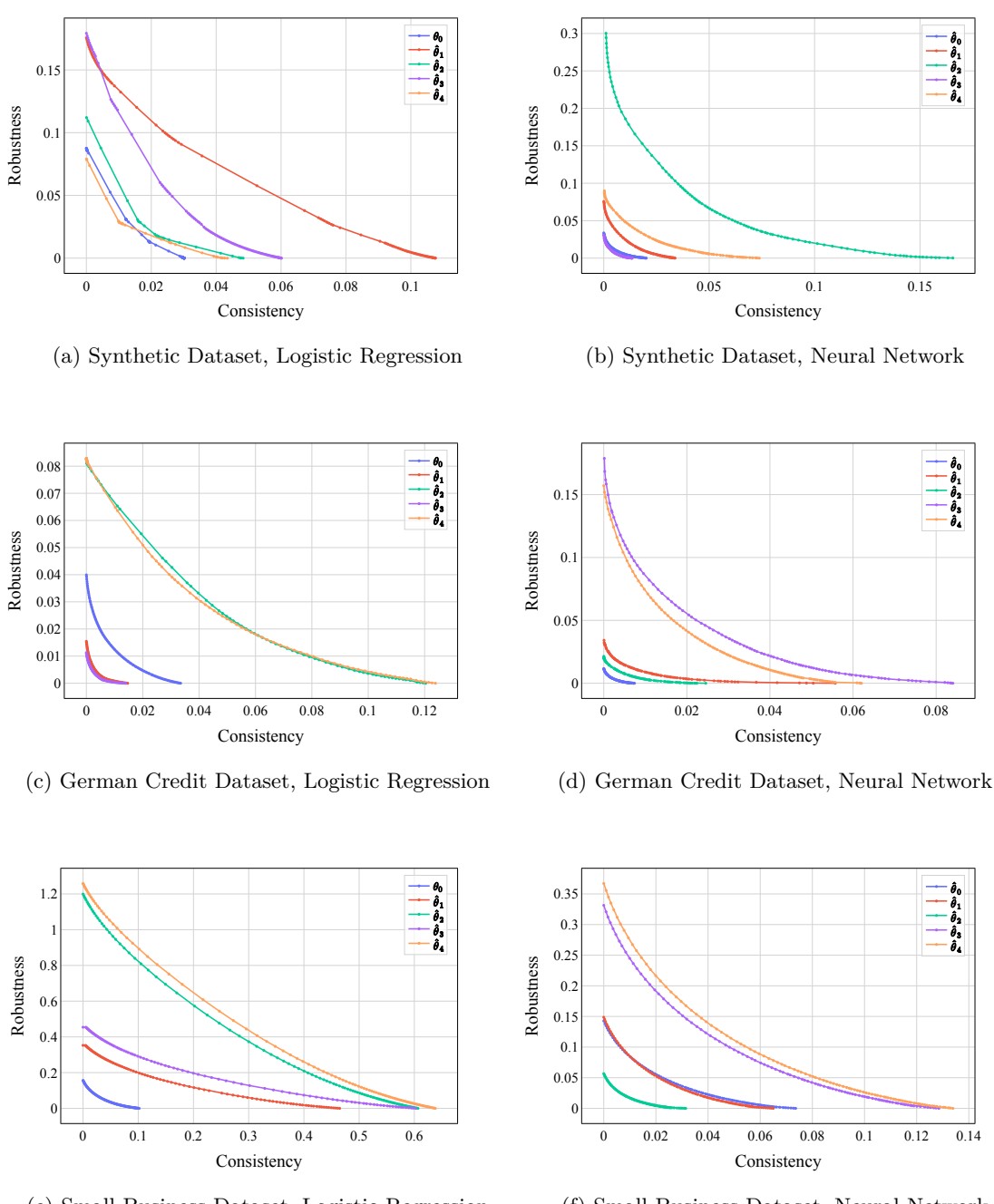

(a) Synthetic Dataset, Logistic Regression

(b) Synthetic Dataset, Neural Network

(c) German Credit Dataset, Logistic Regression

(d) German Credit Dataset, Neural Network

(e) Small Business Dataset, Logistic Regression

(f) Small Business Dataset, Neural Network

Figure 6: The trade-off between robustness and consistency for $\alpha = 0.1$: logistic regression (left) and neural network (right). Rows correspond to datasets: synthetic (top), German (middle), and Small Business (bottom). In each subfigure, each curve shows the trade-off for different predictions as mentioned in the legend.

## C.5  Baselines for Smoothness Results

In this section, we revisit the smoothness results in Section 4.1 by including a baseline. For any $\beta$, the optimization problem in Equation 7 can be solved by the type of gradient-based techniques that are common

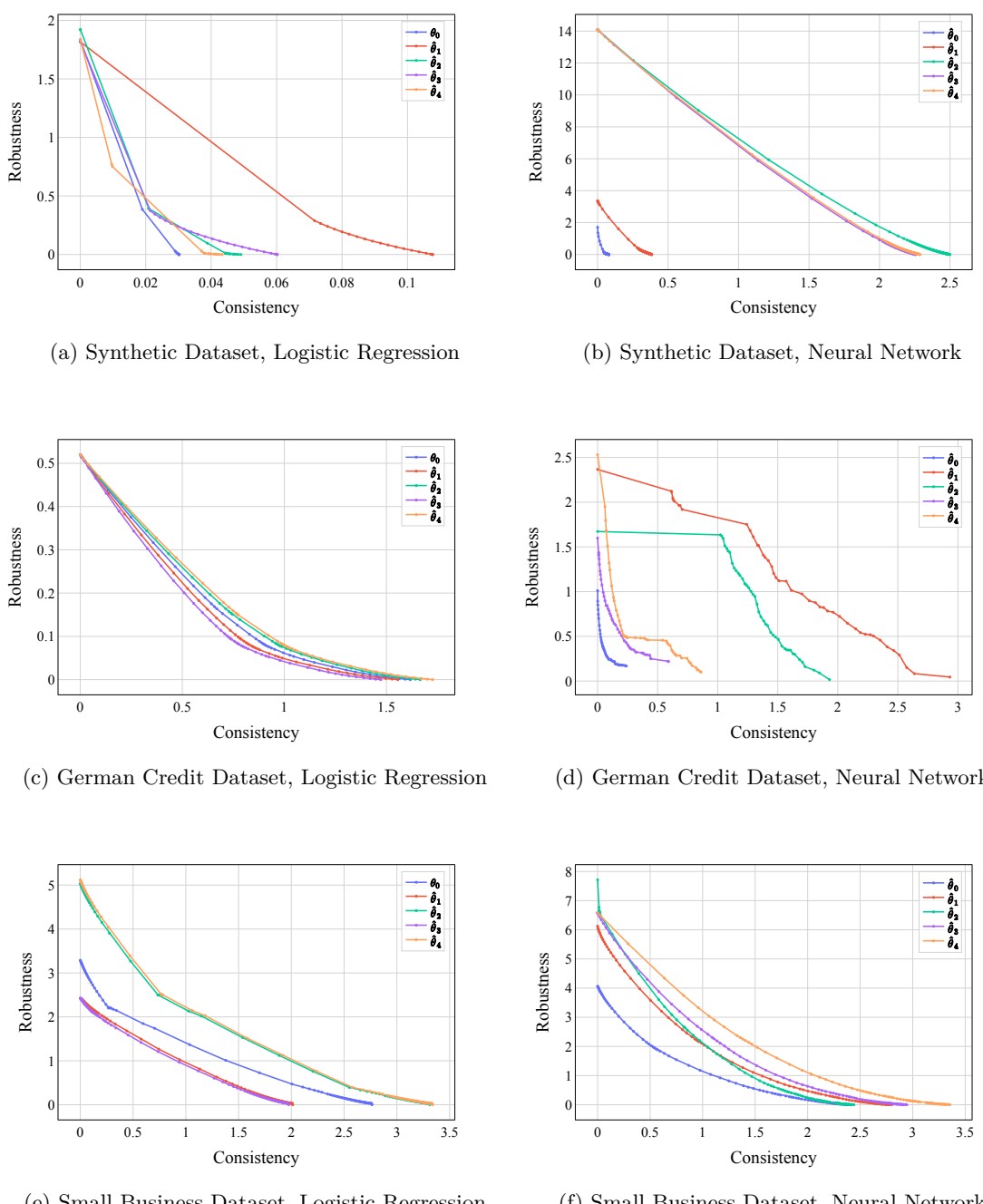

Figure 7: The trade-off between robustness and consistency for $\alpha = 1$: logistic regression (left) and neural network (right). Rows correspond to datasets: synthetic (top), German (middle), and Small Business (bottom). In each subfigure, each curve shows the trade-off for different predictions as mentioned in the legend.

in adversarial training (Madry et al., 2018). Since ROAR is one such algorithm, we can modify it by changing its objective function and use it as a baseline to Algorithm 1.

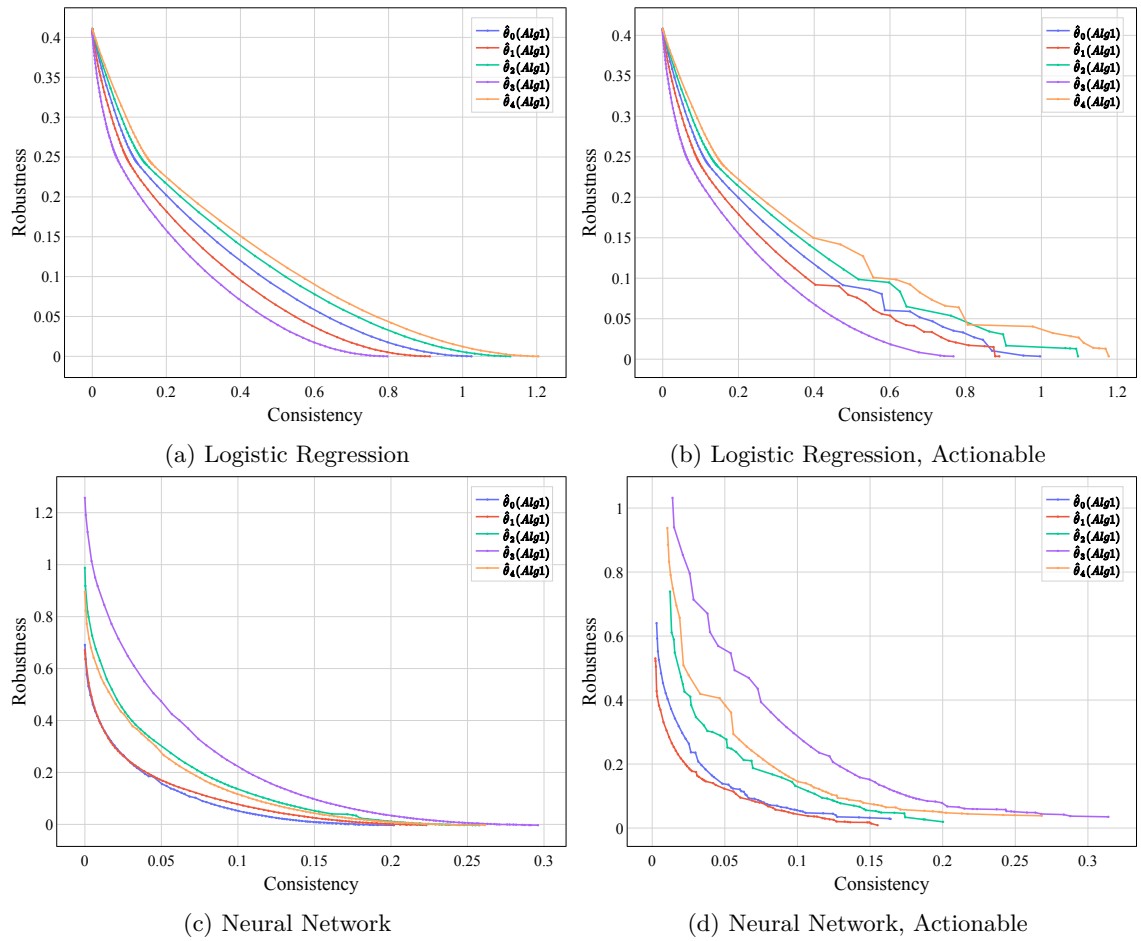

(a) Logistic Regression

(b) Logistic Regression, Actionable

(c) Neural Network

(d) Neural Network, Actionable

Figure 8: The trade-off between robustness and consistency for $\alpha = 0.5$ for German Credit Dataset: logistic regression (top) and neural network (bottom). The left panel corresponds to recourses provided by Algorithm 1, while the right panel corresponds to recourses provided to only actionable features and post-processed for categorical features. In each subfigure, each curve shows the trade-off for different predictions.

The results are presented in Figure 9. In each subfigure, the left panel represents the smoothness of Algorithm 1 (exactly as it is depicted in Figure 2) and the right panel depicts the smoothness of ROAR. There are a couple of interesting observations: First of all, similar to our algorithm, following $\hat{\theta}^*$ at $\beta = 0$ will result in the lowest smoothness value, though compared to following other predictions though unlike our algorithm, the smoothness in this situation can be higher than 0. On the other hand, at the other extreme, the smoothness of ROAR converges for different predictions since, in this case, the prediction is ignored. Moreover, the smoothness of ROAR for each prediction also exhibits non-monotone behavior as a function of $\beta$. However, the $\beta$ at which the non-monotonicity starts to emerge is different for ROAR compared to our algorithm. Finally, we generally observe that the smoothness of ROAR using the same prediction and $\beta$ is generally much worse than our algorithm (note that the Y axis has different scales in the left and right panels).

## C.6 Computing Robust Recourse

In this section, we repeat the experiments in Section 4.2 but also include comparisons with RBR (Nguyen et al., 2022). Figure 10 depicts the trade-off between worst-case validity and cost for all datasets (rows) and models (columns). Since RBR does not have the same parameters as our algorithm and ROAR, the trade-off for RBR is obtained by replicating their experiments and setting the ambiguity sizes to $\epsilon_0, \epsilon_1 \in [0, 1]$ with increments of 0.5, and the maximum recourse cost $\delta = \|x_0 - x_r\|_1 + \delta_+$ to $\delta_+ \in [0, 1]$ with increments of 0.2.

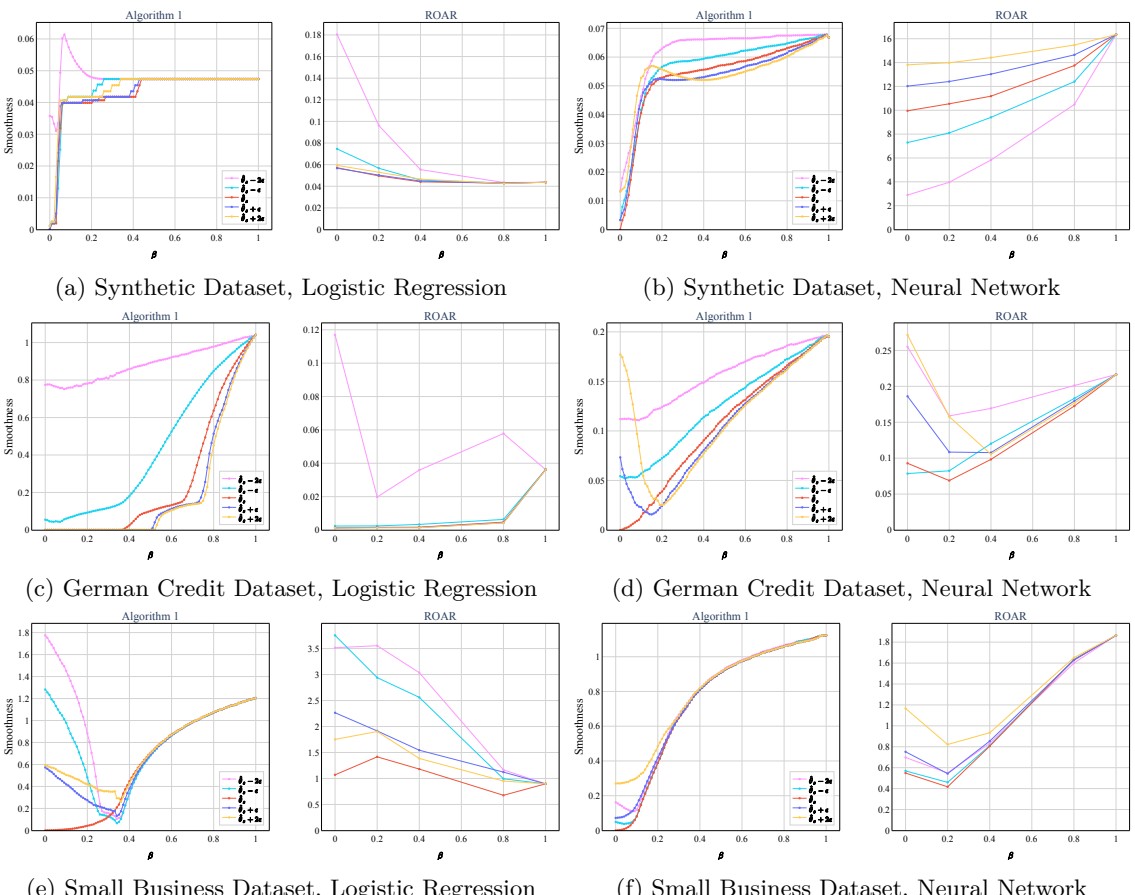

Figure 9: The smoothness analysis for predictions with different accuracies: rows and columns correspond to different datasets and models as indicated in the sub-caption. In each subfigure, each curve corresponds to a different prediction and tracks the smoothness as a function of $\beta$ for the given prediction. In each subfigure, the left corresponds to Algorithm 1 and the right corresponds to ROAR.

For neural network models, the fidelity of LIME approximation for Synthetic, Small Business, and German datasets is 0.93, 0.54, and 0.02 when averaged over instances where recourse is provided. These fidelity values represent a wide range for the approximation quality of LIME. We see, in the right panel of Figure 10, that our algorithm performs similarly to or better than RBR when fidelity is relatively reasonable, and only when fidelity is extremely low (German dataset) does its performance become dominated by RBR. For logistic regression models, in the left panel of Figure 10, our algorithm generally finds very high validity, while this is not the case for RBR, which sometimes achieves validity even smaller than ROAR. We also observe that, to achieve lower or moderate validity values, $\lambda$ needs to be set higher than 0.3 for our algorithm.

## C.7 Experiments on Larger Datasets

In this section, we provide experimental results for a much larger dataset (both in terms of the number of instances and number of features) compared to the datasets in Section 4. The running time of our algorithm scales linearly with the number of instances for which recourse is provided. For each instance, the running time of our algorithm grows linearly in the number of features since the minimization problem in Line 13 of our algorithm can be solved analytically. For non-linear models, the cost of approximating the model with a linear function should be added to the price per instance.

We use the ACSIncome-CA (Ding et al., 2021) dataset for experiments in this section. This dataset originally consisted of 195,665 data points and 10 features, 7 of which are categorical and have been one-hot encoded.

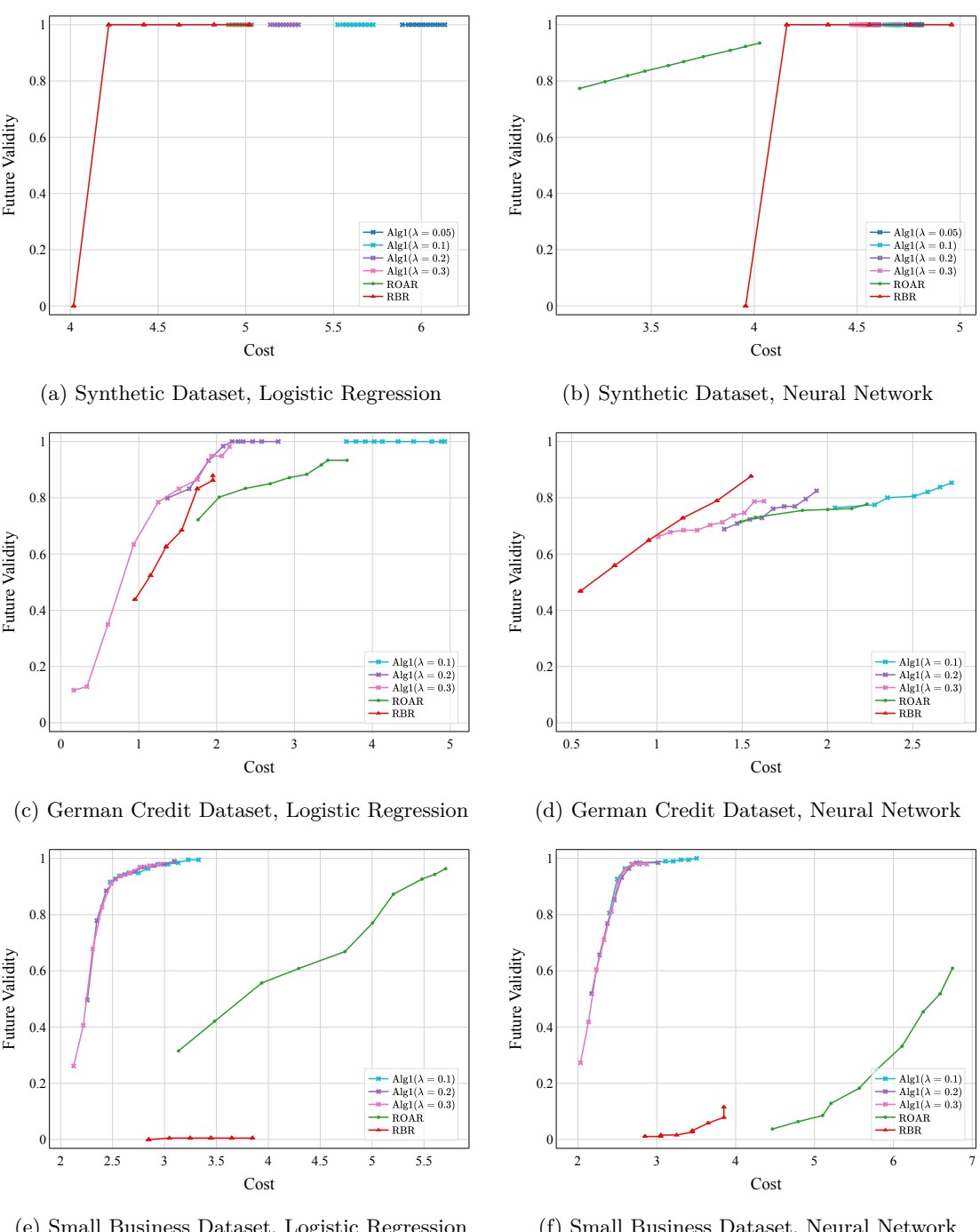

(a) Synthetic Dataset, Logistic Regression  (b) Synthetic Dataset, Neural Network

(c) German Credit Dataset, Logistic Regression  (d) German Credit Dataset, Neural Network

(e) Small Business Dataset, Logistic Regression  (f) Small Business Dataset, Neural Network

Figure 10: The trade-off between future validity and cost: rows and columns correspond to different datasets and models as indicated in the sub-caption. In each subfigure, curves show the trade-off for different algorithms specified in the legend.

However, to lower the runtime, we subsampled the dataset to include 50,000 data points and removed the categorical feature "occupation (OCCP)", as it contains more than 500 different occupations. This resulted in more than 250 features after one-hot encoding.

Figure 11 depicts the trade-off between the cost and validity of recourse for both logistic regression and neural network models. The choices of parameters used for results in Figure 11 are the same as the results for Figure 10 in Section 4.2. We observe that even in a dataset with a much larger number of features, Algorithm 1 can generate recourses with high validity, especially for logistic regression models. Similar to Figure 10, achieving very high validity comes at a cost of higher implementation cost, which is higher than the cost required for smaller datasets.

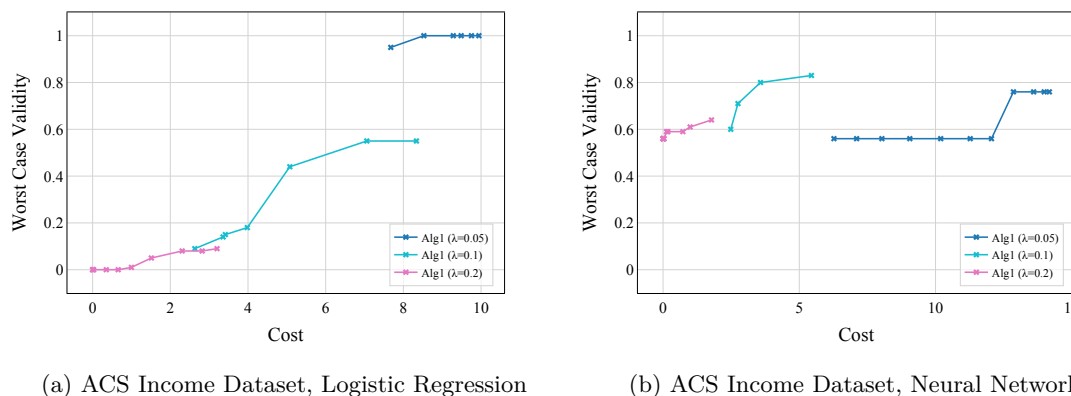

(a) ACS Income Dataset, Logistic Regression      (b) ACS Income Dataset, Neural Network

Figure 11: The trade-off between the worst-case validity and the cost of recourse. The left panel is for the logistic regression, while the right panel is for a neural network for the ACS Income dataset. In each subfigure, each curve shows the trade-off for different methods as mentioned in the legend.

