# OpenReview forum: "Learning-Augmented Robust Algorithmic Recourse"
_TMLR — Accepted by TMLR_

### Review · Reviewer_GgHh · 2026-02-10

**Summary Of Contributions:**

The paper studies learning-augmented robust algorithmic recourse, motivated by the fact that deployed models are frequently updated, so a recourse computed for today’s model may fail tomorrow. The authors assume the recourse designer has access to an unreliable prediction of the future model parameters and propose to optimize a robustness-consistency tradeoff: near-optimal performance when the prediction is accurate (consistency) while retaining worst-case guarantees when it is not (robustness).

Technically, they define robustness/consistency measures and optimize their convex combination, giving Algorithm 1 to compute tradeoff recourses. They further prove optimality for generalized linear models at the two extremes (robust-only/consistent-only), and support the approach with experiments on synthetic and real datasets comparing to ROAR.

**Audience:**

Yes

**Audience Explanation:**

The framing (learning-augmented recourse) feels natural and timely, and the robustness/consistency definitions are clean and easy to interpret. Empirically, the method often Pareto dominates ROAR in robustness-consistency and validity-cost tradeoffs, and appears substantially faster in the reported implementations.

**Broader Impact Concerns:**

No direct ethical red flags.

**Claims And Evidence:**

No

**Claims Explanation:**

I am only selecting no, so I can paste what I feel are the weaknesses of the paper.

The strongest theoretical guarantee is for $\beta \in \left\lbrace{ 0,1}\right\rbrace$ (the extremes), whereas much of the motivation is about the continuous tradeoff for intermediate $\beta$'s.

For non-linear models, the approach relies on local linearization (e.g. LIME), which raises questions about how approximation error interacts with robustness claims in practice.

Finally, the experimental “predictions” appear to be constructed via controlled perturbations; it would be helpful to better justify how these map to realistic model-update mechanisms.

Theorem 3.3 is a really nice one.

**Requested Changes:**

**Q1 (Theory/scope).** Can the authors clarify whether anything can be said for intermediate $\beta \in (0,1)$ beyond empirical performance? Even partial results (e.g. conditions under which the objective becomes well behaved, or approximation guarantees) would strengthen the story.

**Q2 (Nonlinear setting).** Since the method for neural nets depends on a local surrogate, can the paper add an explicit discussion (or diagnostic) of surrogate fidelity and how it affects robustness/consistency measurements? An ablation varying locality/explanation quality would be valuable.

**Q3 (Prediction realism/$\alpha$ selection).** The framework assumes a known radius $\alpha$ for model drift and uses handcrafted predictions in experiments. Could the authors add guidance on how $\alpha$ and predictions might be estimated in practice (or at least a sensitivity study/failure modes)?

---

> ### Author Response · Authors · 2026-03-11
>
> We thank the reviewer for the feedback. Please see our answers below. All the changes in the text are colored in red in the revised version.
>
> **Q1:** Although the interesting cases of robustness ($\beta=1$) and consistency ($\beta=0$) provide sufficient structure for us to be able to design an optimal algorithm, values of $\beta \in (0,1)$ give rise to a much more complicated optimization problem for one to hope to achieve theoretical optimality. Specifically, fully optimizing a convex combination of the R(.) function and the C(.) function heavily depends on the relative contribution of each of these functions to the overall objective. We also do not see any direct implications regarding approximation guarantees that one could achieve for these values of $\beta$. We therefore focus on detailed empirical evaluation in Section 4.1 instead, demonstrating that our optimal algorithm for $\beta=1$ performs well in this intermediate regime. It is important to emphasize that prior work in this area provided no theoretical guarantees for the $\beta=1$ case and worked with heuristics instead, so we strongly believe that $\beta=1$ alone is of significant interest.
>
> **Q2:** This is a great question. LIME approximation can indeed be unreliable, and we did perform experiments to account for this. In Section C6, we compared the performance of our algorithm to RBR, which is designed to address the same shortcoming that the reviewer mentioned. In these experiments, the fidelity of LIME approximation of the neural network for Synthetic, SBA, and German datasets is 0.93, 0.54, and 0.02 when averaged over instances where recourse is provided (we added these numbers to Appendix C6). The results are summarized in the right panel of Figure 10 (Appendix C6). We see that our algorithm performs similarly to or better than RBR when fidelity is relatively reasonable, and only when fidelity is extremely low (German) does its performance become dominated by RBR. And even in those cases, our algorithm still achieves a high validity (despite a higher cost). We updated the main text (last paragraph of Section 4.2) to provide more information about these observations.
>
> **Q3:** $\alpha$ is a user-defined parameter, and we provided sensitivity analysis for $\alpha$ in Appendix C3, demonstrating the effectiveness of our approach for various $\alpha$ values.
>
> As for the predictions, in the learning-augmented setting, it is assumed that the prediction is given. Hence, we experimented with natural ways to relate such a prediction to the true parameters (i.e., by adding different degrees of noise to the true parameters). In the discussion, we highlighted how such a prediction might be obtained in specific settings: “A natural way to compute such a prediction, in practice, is through performativity (Perdomo et al., 2020). For example, the implementation of recourse can cause the distribution shift, and the designer can use this knowledge to form a prediction for the anticipated future model (König et al., 2025). However, this prediction can be imperfect since individuals might not exactly implement the recourse (Fonseca et al., 2023).” We would be happy to expand the discussion.

---

### Review · Reviewer_9mcs · 2026-03-04

**Summary Of Contributions:**

The authors propose a novel problem setting of learning-augmented robuist algorithmic recourse. In algorithmic recourse, an instance receives a negative label, and the goal is to make a least-cost change to that instance's features so that the label becomes positive. In the robust setting, the model may change arbitrarily within an epsilon-ball in the L-infinity sense. In the novel learning-augmented setting, the algorithm is given access to a (possibly inaccurate) prediction of the updated decision-making model. The goal is to optimally trade off between robustness (handling cases where the predicted model is highly inaccurate) and consistency (leveraging high-quality predictions for the updated model). The authors propose an algorithm that finds globally optimal solutions for any point in the convex combination of robustness and consistency, yielding also the first globally optimal algorithm for the pure robustness case.

######## Strengths ########

- The exposition of the manuscript throughout sections 1, 2 and 3 (not including 3.1) is of very high quality.
    - One smaller comment is that part of the problem setting seemed somewhat unclear at first glance to me and required a careful re-read. In particular, what defines "a recourse" in the sense that can be valid in the original model but invalid in the updated model?
        - Initially, I thought that the authors implied that the minimizer of Equation 1 was stored in some (parametric) function, and that the recourse framework did not allow us to solve the minimization in (1) for each instance, but instead we find a solution to (1) that generalizes across x's. My first understanding was then that each new instance/applicant could face an updated model.
        - Instead, I believe that the point is that once a minimizer x' is found for an individual instance, we want that minimizer to remain valid (for that instance) even if the model changes. So we focus on one individual applicant and how changes to the model affect this individual applicant's recourse.
        - This may be well known in the literature, and the authors' phrasing does not directly lead to my first interpretation, but it may be worth making the explanation more explicit to ensure that my (I think correct) second understanding is what readers take away.
- The coverage of related works appears fair, comprehensive, and informative. While I'm not familiar with the literature (and so I cannot validate that all closely related literature is discussed), I found that the description gave me a useful mental model for how this work relates to other areas including explainability and adversarial robustness.


######## Weaknesses ########

- Sec 3.1 includes only minimal technical detail, relegating most of the technical algorithmic and theoretical contributions to an appendix.
    - This section provides no intuition at all for what Algorithm 1, just the mechanics. Even going through the appendix, the only way to get a sense of why this algorithm works is to read the proof. I encourage the authors to include a brief, high-level explanation of the key ideas that make it possible for this algorithm to work (e.g., the intuition behind Observation B.1, the key insight that theta can only move within an epsilon ball in the L-infinity sense which allows the Algorithm to operate on individual dimensions independently)
- Some technical details in Appendix B are somewhat incomplete or inaccurate:
    - The "proof by example" for non-convexity is not very convincing. Does this generally happen for some class of losses $\ell$? Is the $\text{sgn}$ function the culprit of

    - Appendix B2 states that Eq 7 and 5 are the same, but this is technically inaccurate, since 7 is the minimization problem and 5 is the objective. The connection between Equation 6 and 3 has the same issue. Perhaps more importantly, the authors never explain or prove why it is sufficient to show that Algo 4 optimizes (3), if (7) is a convex combination of two objectives similar to (3). I believe that the authors mean to convey that the proof for Algo 1 would be very similar in the case of Eq. (7), but I don't necessarily believe that there is a direct implication (as would be the case if instead we proved that Eq. 3 was convex).

**Additional Comments:**

The following points are provided as feedback to hopefully help better shape the submitted manuscript, but will not impact my recommendation in a major way.

The proofs in Appendix B seem to follow from first principles, rather than rely on heavy mathematical machinery. This makes it so that the paper could be digested even by people outside of the research area. For that, it would be useful to include at least one paragraph, ideally in the main paper, explaining at a high level the intuition behind the algorithm and proof. My thoughts on this include:
    - One key piece is the assumption that the adversary can only move within an l-infinity ball of size alpha, which makes it possible to reason about dimensions independently
    - Another key piece is assuming that the cost is convex and decreases with the inner product of theta and x
        - This is the case for linear models where the cost measures how negative a prediction is (the user/applicant wants to flip the label to positive, so negativeness is penalized)

Typos/style/grammar
- Intro: "... even in the worst-case" --> "... even in the worst case"
- Throughout: right quotation marks should be two separate ' symbols
- Sec 4: "... three-level neural network..." --> "... three-layer neural network..."
- Appendix B.1: should be $\text{sgn}$ as in Sec 3.1

**Audience:**

Yes

**Audience Explanation:**

The contents of the manuscript appear to be of interest to subsets of the theory community.

**Claims And Evidence:**

No

**Claims Explanation:**

Regarding the paper's claims, my main two concerns are explained in the weaknesses above:
1. The key theoretical claims are supported exclusively by Appendix B, with only bare-minimum descriptions in the main paper.
2. The bridge between the optimization to Eq. (3) and the optimization to Eq. (7) is missing.

I expect that these can be addressed in the author discussion period and I would be happy to change my answer to "yes".

**Requested Changes:**

See weaknesses and "claims" box above.

---

> ### Author Response · Authors · 2026-03-11
>
> We thank the reviewer for the feedback. Please see our answers below. All the changes in the text are colored in red in the revised version.
>
> **The key theoretical claims are in Appendix B**: We agree with the reviewer that we should provide additional information regarding the proof of our main theoretical result in the main body of the paper. We have revisited the description of the algorithm aiming to further simplify and clarify it, and we have also added a sketch of the proof for Theorem 3.3 in the main body, which incorporates the two key pieces pointed out by the reviewer.
>
> **The bridge between Equation (3) and Equation (7):** First note that the RHS of Equations (3) and (5) are both for computing robust recourse. More specifically, Equation (5) is the same as Equation (3) minus the extra term. This extra normalization term ensures that the optimal solution to Equation (5) has an objective value of 0.
>
> Second, Equations (4) and (6) are for consistency, and again, the only difference in the RHS is that the additional normalization term in Equation (5) ensures that the optimal solution has an objective value of 0.
>
> Finally, Equation (7) allows us to study the trade-off between robustness and consistency by optimizing for $\beta$ times Equation (5) and $(1-\beta)$ times Equation (6). $\beta=1$ recovers the robustness objective and $\beta=0$ recovers the consistency objective. The normalization for both equations comes in handy here because it makes sure the scales of the two terms in Equation (7) are comparable.
>
> Based on this explanation, to solve Equation (7) for $\beta=1$ it suffices to solve Equation (3). And to solve Equation (7) with $\beta=0$, it suffices to solve Equation (4), as in both cases we can ignore the normalization term, which does not change the optimal recourse. However, Equation (4) is a special case of Equation (3) (replace $\theta_0$ with $\hat{\theta}$ and set $\alpha$ to 0). Therefore, to prove Equation (7) can be solved optimally by our algorithm for $\beta\in \{0,1\}$, it suffices to show that the algorithm can solve Equation (3) optimally.
>
> **Proof by example for non-convexity:** We added the formal statement for non-convexity in Proposition B1 in Appendix B1. At a high-level, to show that our robust recourse formulation is, in general, non-convex (i.e., if we solve for $\theta$ as a function of any fixed $\alpha$ and $x$, we show that this leads to a non-convex optimization problem in $x$). Therefore, to show non-convexity, we only need to provide an instance of our problem that satisfies all of our assumptions and show that the instance will result in a non-convex optimization problem. The proof provides one such instance (so an example suffices here).
>
> **Instance-level formulation of recourse:** It is well-known (see e.g. Upadhyay et al., 2021 or Nguyen et al., 2022) that the recourse is computed at an instance level and robustness requires this instance-level recourse to be valid even if the model changes slightly.
>
> **Typos:** Thanks for pointing these out. They are now fixed.

---

> > ### Comment · Reviewer_9mcs · 2026-03-17
> > **Follow-up**
> >
> > Thank you for your clarification. Reading the other reviews and the authors' comments, I have no further questions. Two final comments:
> > 1. As mentioned in other reviews, the contributions would be better framed if the authors clarified from the beginning that the theoretical results apply only in the extreme cases of $\beta\in\{0,1\}$.
> > 2. Regarding the instance-level recourse formulation, I agree with the authors. Nonetheless, because this is a well-written, beginner-friendly manuscript, I encourage the authors to make this description explicit to further aid in introducing newcomers to the field.

---

> > > ### Author Response · Authors · 2026-03-17
> > >
> > > Thank you. We agree with these two final comments and incorporate them in the final version.

---

### Review · Reviewer_wwXX · 2026-03-06

**Summary Of Contributions:**

This paper studies learning augmented recourse. In particular, given an original model $\theta_0$, and an uncertainty set $\Theta_{\alpha}$ around this model, the goal is to find a decision given some prediction $\hat{\theta}$. The goal is to produce recourse that is cheap when the prediction is accurate (consistency) while remaining safe under adversarial model shifts (robustness). The paper formalizes robustness and consistency (relative to the robust-optimal and prediction-optimal solutions) and proposes an optimization objective that trades them off via a parameter $\beta$.

Strengths:

1. The paper is clear about its goal and provided rationale behind its chosen objective.

2. The algorithm is simple to implement.

3. The empirical results are strong.

Weaknesses:

1. In the learning-augmented algorithms literature, one typically expects formal robustness/consistency guarantees (e.g., bounds that smoothly degrade with prediction error, competitive ratios, or instance-wise guarantees). Here, the core learning-augmented story is: define a tradeoff objective and then empirically study “smoothness”. That is useful, but it is weaker than what the framing suggests.

2. The paper hinges on the fact that the robustness would be conservative if someone has some good prediction about the future model $\hat{\theta}$. Thus, it tries to hedge between these two. However, solving this combination might not be easy, given that some nested loop involved. However, there are a few other choices compared to the min-max robustness. For example, one can consider CVaR or quantile optimization. It is not clear whether the required formulation is needed.

3. The paper relies on $\alpha$, the uncertainty around the model parameter; however, it is not clear how in practice, one can achieve that. Further, how can one tune the $\beta$ parameter?

4. In Theorem 3.3, the result only holds for $\beta \in$ { 0,1}. It does not hold for other values of $\beta$ between $0$ and $1$ which makes the results weaker.

**Audience:**

Yes

**Audience Explanation:**

It will be of interest to the community.

**Claims And Evidence:**

No

**Claims Explanation:**

While the paper has an interesting idea, the claims are not supported yet. In particular, the reviewer has concerns about the theoretical results, the main problem formulation, and the hyper-parameter tuning.

**Requested Changes:**

Please answer the weaknesses.

The paper mentioned about non-convexity, however, it is a min-max problem, so it is convex as long as the loss function is convex. Can the authors comment on that?

Instead of max, if someone considers a smoother function like soft-max, would it be more stable? Can the authors compare with that baseline?

---

> ### Author Response · Authors · 2026-03-11
>
> We thank the reviewer for the feedback. Please see our answers below. All the changes in the text are colored in red in the revised version.
>
> **Convexity of the problem:** Similar to bilevel optimization (e.g., Stackelberg games, adversarial training, etc.), when the loss functions, models, and cost functions are convex, the problem is convex for any fixed $\theta$. Similarly, for any fixed $x$, the problem is concave in $\theta$. However, this convexity/concavity *does not* make the problem easy to solve, and most often only local or stable solutions are guaranteed (see e.g., https://proceedings.neurips.cc/paper_files/paper/2023/hash/c981fd12b1d5703f19bd8289da9fc996-Abstract-Conference.html).
>
> For the specific case of our formulation, where we can solve $\theta$ as a function of any fixed $\alpha$ and $x$, we show that this leads to a non-convex optimization problem in $x$.
>
> **Learning-augmented framework:** The learning-augmented framework studies the trade-off between robustness (performance in an adversarial setting) and consistency (performance given accurate predictions). The notion of robustness and consistency, as well as the type of prediction, depends on the setting at hand. In the setting that we consider, the notion of robustness that prior work has focused on allows for the model to shift adversarially within a neighborhood determined by parameter $\alpha$, and the performance of algorithms facing this adversarial shift is then evaluated experimentally. In our work, we use that same notion of robustness and empirically study the trade-off between this robustness and the natural notion of consistency. Regarding the “tradeoff objective” that the reviewer is referring to, we use that as a guide for our algorithm, aiming to optimize different trade-offs of robustness and consistency, but our evaluation of this algorithm is based on robustness and consistency, as is standard in the learning-augmented literature (see Figure 1). In fact, apart from robustness and consistency trade-offs, we even evaluate the smoothness of our algorithms as a function of the prediction error (Figure 2), which goes beyond the robustness and consistency measures that a lot of prior work in the learning-augmented literature is restricted to.
>
> **concerns about the theoretical results:** We are puzzled by the reviewer’s statement regarding their concerns about the theoretical results. We provide an algorithm, and we formally prove that this algorithm can be used to achieve optimal robustness or optimal consistency. This is in contrast to prior work in the robust recourse recommendations that used heuristics instead. Is the reviewer suggesting that our statement/algorithm is incorrect?
>
> **CVaR or quantile optimization:** As we stated in the conclusion ``our focus was on the empirical evaluation of our algorithms and, in line with prior robust recourse work.” Utilizing other robustness frameworks is outside of the scope of this paper; hence, we adjusted the conclusion to mention the reviewer’s suggestion.
>
> **Softmax instead of max:** We do not understand what stable means in this comment. As far as we are aware, softmax is not used in any prior robust recourse framework, so the justification for this baseline is unclear to us.
>
> **Choice of $\alpha$:** $\alpha$ is a user-defined parameter and larger values of $\alpha$ correspond to even more demanding notions of robustness, as they allow the model to shift adversarially within an even bigger “neighborhood”. The user can choose $\alpha$ depending on the level of change that they wish to protect against in terms of robustness, and we provided sensitivity analysis for $\alpha$ in Appendix C3, demonstrating the effectiveness of our approach for various $\alpha$ values.

---

> > ### Comment · Reviewer_wwXX · 2026-03-14
> > **Clarifying the comments**
> >
> > Dear Authors,
> >
> > I apologize if my questions have annoyed you (which seems to be the case from the tone of the response). However, I want to make the paper better, and want to understand the contributions better. I have tried to explain my points better.
> >
> > My point was that if you see the statement of Theorem 3.3, it only covers two extreme scenarios where $\beta=0$, and $\beta=1$. The proof does not extend to other cases, where $\beta \in (0,1)$, which significantly weakens the claim.
> >
> > Regarding the convexity, I got the point. However, the question is if one can convexify, then how bad would be the approximation be?
> >
> > Regarding the baselines, we believe that the paper can be strengthened if it can show that even with applying smoothness like softmax (if it is easy to do that), the proposed baseline is unstable or has a bad performance compared to the proposed approach.

---

> > > ### Author Response · Authors · 2026-03-16
> > >
> > > Thank you for engaging with us during the discussion period. Our sincere apologies if our response read that way. This was not our intention, and we just want to make sure we understand your concerns clearly. Please see our responses below.
> > >
> > > **Theorem 3.3:** Based on your follow-up comment, our understanding is that you do not have any concerns regarding the correctness of the theorem and its proof. Instead, the concern is that our theoretical optimality results focus on the case of $\beta=1$ and $\beta=0$. We would like to emphasize that prior work (e.g., the well-cited paper of Upadhyay et al.) was restricted to just $\beta =1$, and their evaluation did not just lack theoretical optimality results for it; it actually used sub-optimal heuristics for solving this problem. Apart from providing a thorough experimental evaluation of our improved algorithms and examining the achievable trade-offs between robustness and consistency (which is the main part of our contribution), we also formally prove the theoretical optimality of our algorithms for $\beta=1$ (as well as $\beta=0$). Given this context, suggesting that our theoretical results are *insufficient* or *not strong enough* seems somewhat unreasonable.
> > >
> > > **Convexity and approximation ratio:** In typical bi-level optimization problems (as is the case for us), the objective function is convex for each of the variables when fixing the action of the other variable. In our problem, we can solve for the best response of the adversarial model for any fixed recourse (and alpha) and show the resulting objective is non-convex. We are unsure how to convexify this objective. Simply replacing the sign with a convex proxy might be insufficient here since the sign is getting multiplied by other terms. Moreover, even if this convexification can be done, we are unsure how it can be used to prove an approximation guarantee. We would appreciate it if the reviewer could provide any suggestions on what exactly they have in mind
> > >
> > > **Softmax baseline:** The most natural baselines to us are the prior approaches (ROAR and RBR) that solve the same (or almost the same) problem with different algorithms. We do not see a justification for including a baseline like Softmax, since it solves a different problem. Moreover, it is unclear to us how to implement such a baseline since the set of models in $\Theta_{\alpha}$ is not finite. We would appreciate it if the reviewer could explain this point in more detail and also explain what they mean by instability.

---

### Decision · Action_Editor_jGjv · 2026-04-16

**Recommendation:** Accept with minor revision

**Additional Comments:**

The paper is generally in good shape after revision. However, one reviewer suggested clarifying the scope of the theoretical results in the introduction, and I did not see corresponding changes in that section. Explicitly stating this in the introduction would help readers understand the scope and limitations of the results at an early stage.

**Audience:**

Yes

**Audience Explanation:**

Yes, all reviewers agreed that the paper would be of interest to at least part of the TMLR audience. In particular, multiple reviewers explicitly noted that the work would appeal to theory-oriented readers, especially those interested in learning-augmented algorithms and robust optimization.

**Claims And Evidence:**

Yes

**Claims Explanation:**

This submission presents a novel formulation of learning-augmented robust algorithmic recourse. It develops an efficient algorithm and a theoretical optimality result for generalized linear models in the extreme cases of $\beta=0$ and $\beta=1$. In the original submission, while the theoretical results appear correct, several reviewers noted that the scope of the optimality result was not clearly presented, and key arguments were relegated to the appendix. The empirical results are strong but do not verify theoretical claims. In addition, the intuition behind the proposed algorithm was not sufficiently explained.

In the revised version, the authors have adequately addressed most of these concerns. As a result, the claims are now appropriately scoped and supported by convincing and sufficiently clear evidence. One remaining minor issue is that, the scope of the theoretical results is still not prominently emphasized in the introduction, as suggested by one reviewer.